# Laboratory earthquakes decipher control and stability of rupture speeds

Peng Dong [1], Kaiwen Xia [1,2,3] ✉, Ying Xu [3], Derek Elsworth[4] & Jean-Paul Ampuero [5]

Earthquakes are destructive natural hazards with damage capacity dictated by rupture speeds. Traditional dynamic rupture models predict that earthquake ruptures gradually accelerate to the Rayleigh wave speed with some of them further jumping to stable supershear speeds above the Eshelby speed (~$\sqrt{2}$ times S wave speed). However, the 2018 $M_w$ 7.5 Palu earthquake, among several others, significantly challenges such a viewpoint. Here we generate spontaneous shear ruptures on laboratory faults to confirm that ruptures can indeed attain steady subRayleigh or supershear propagation speeds immediately following nucleation. A self-similar analysis of dynamic rupture confirms our observation, leading to a simple model where the rupture speed is uniquely dependent on a driving load. Our results reproduce and explain a number of enigmatic field observations on earthquake speeds, including the existence of stable subEshelby supershear ruptures, early onset of supershear ruptures, and the correlation between the rupture speed and the driving load.

The severity of seismic damage is significantly affected by rupture speed ($V_r$)[1], which spans from a few hectometers per second to the P wave speed ($C_P$) of the host rock[2,3], except for the forbidden zone between the Rayleigh wave speed ($C_R$) and the S wave speed ($C_S$)[4]. Particularly, supershear earthquakes with rupture speeds exceeding $C_S$[5–9] can induce increased damage due to the passage of the Mach wave[1,10,11]. The control on rupture speed is thus crucial for understanding earthquakes and the attendant assessment of seismic hazards.

Although seismological observations and inverse analyses have provided increasingly well-resolved source parameters for natural earthquakes, key questions remain in linking observations to fundamental theories of earthquake ruptures. A particularly mystifying aspect of earthquake rupture is the mechanism for slow-supershear ruptures propagating at theoretically unstable speeds between $C_S$ and the Eshelby speed $C_E$ ($\sim \sqrt{2}C_S$)[12,13], as in the 2018 $M_w$ 7.5 Palu earthquake[14], the 1999 $M_w$ 7.1 Düzce earthquake[15], the 1979 $M_w$ 6.5 Imperial Valley earthquake[16], and the 2017 $M_w$ 7.7 Komandorski

earthquake[17]. To reconcile this apparent contradiction, several models have been proposed, such as the low-velocity fault zone model[16,18], faults with geometric complexity[19], and mixed-mode ruptures involving oblique slip[20]. However, these models all require special conditions and are incapable of interpreting stable subRayleigh speeds.

Additionally, how a rupture along one-dimensional homogeneous faults attains a supershear speed remains an open question. Theoretical studies[21,22] and laboratory experiments[23] have shown that such a rupture often gradually accelerates to Rayleigh speed and that some subRayleigh ruptures can jump over the forbidden zone to a supershear speed through the Burridge-Andrews mechanism - where the jump to supershear speeds is achieved through the growth of a separate rupture at the S wave front ahead of the main rupture. Alternatively, rupture can also directly accelerate through this forbidden zone into the supershear region[24]. However, the 2018 $M_w$ 7.5 Palu earthquake attains supershear speed at the onset (or from the first moments that a velocity can be discerned)[14,25], challenging such conventional understanding on supershear nucleation. The controls for

[1]Institute of Geosafety, School of Engineering and Technology, China University of Geosciences, Beijing 100083, China. [2]Department of Civil and Mineral Engineering, University of Toronto, Toronto, ON M5S 1A4, Canada. [3]State Key Laboratory of Hydraulic Engineering Simulation and Safety, School of Civil Engineering, Tianjin University, Tianjin 300072, China. [4]Energy and Mineral Engineering & Geosciences, G3 Center and EMS Energy Institute, Pennsylvania State University, University Park, PA 16802, USA. [5]Géoazur, Université Côte d'Azur, IRD, CNRS, Observatoire de la Côte d'Azur; 250 rue Albert Einstein, 14 Sophia Antipolis, 06560 Valbonne, France. ✉e-mail: kaiwen.xia@utoronto.ca

rupture speeds represent another important question. Some experimental studies propose that rupture velocity depends on the initial stress state[26–28] and static stress drop[29,30], whereas a latest research shows that rupture velocity is controlled by fault roughness[31].

Here we use tightly-constrained laboratory experiments of spontaneous ruptures to address these apparent contradictions between theoretical understanding and seismological observations of earthquake rupture characteristics. Our experiments show that earthquake ruptures attain stable propagation speeds spanning widely from sub-Rayleigh to supershear regions, congruent with field observation of natural earthquakes. The rupture speed is dictated by the driving load that depends on the dynamic stress drop and the normal stress. This driving load controls the strain energy available to drive the rupture propagation. A self-similar rupture model covering speeds from sub-Rayleigh to supershear is developed. The model prediction of rupture speed stability ranges is consistent with both the experimental and the field observations.

## Results

### Stick-slip events on laboratory faults

Experiments are conducted on a square (500 mm × 500 mm) but thin (20 mm) polymethyl methacrylate (PMMA) plate transected by an inclined laboratory fault (Fig. 1a). The roughness of the seismogenic fault is independently controlled (Supplementary Fig. 1; see "Methods" section). The PMMA plates are held together by friction against the biaxial loading (see also Methods and Supplementary Fig. 2). With this loading configuration, the shear stress applied on the fault grows with the differential stress ($\sigma_Y - \sigma_X$). With the continuous increase in the differential stress and thus the shear stress, a series of audible stick-slip

events occur on the fault and produce a measurable static stress drop. These unstable events quasi-periodically release the stress accumulated during the loading process (Fig. 1b). As the horizontal stress ($\sigma_X$) is increased in different experiments, the quasi-periodic nature of the stick-slip events is preserved (Fig. 1c, d) while the instability becomes more violent. This observation is consistent with previous studies on saw-cut laboratory faults on rock[30]. For the same horizontal stress ($\sigma_X$), the peak differential stress at the onset of instability increases with the roughness of the fault. Therefore, we can conclude that the intensity of the stick-slip events is jointly controlled by the stress condition and the fault topography.

For each event, we calculate the global peak shear stress ($\tau_p$) and normal stress ($\sigma_p$) at the onset of instability by resolving $\sigma_X$ and $\sigma_Y$ on the fault plane. Experimental results show that $\tau_p$ and the ratio of global peak shear stress to normal stress ($\tau_p/\sigma_p$) increase with the normal stress and the fault roughness (Fig. 1e). A rougher fault requires higher shear stress to initiate instability under the same normal stress. For all events, the static stress drop, $\triangle\tau_s$, is nearly proportional to $\tau_p$, exhibiting a consistent relative shear stress drop ($\triangle\tau_s/\tau_p$) of ~0.2 (Fig. 1f). These results are qualitatively consistent with existing experimental results[32].

### Spontaneous ruptures of laboratory earthquakes

In this study, ruptures are spontaneously nucleated as stress builds, similar to some existing experiments. Concurrent measurements of the spatiotemporal evolution of fault-parallel slip and the shear strain (Methods and Supplementary Fig. 3) enable the evolution of rupture propagation to be tracked (Fig. 2). These events are all nucleated at random locations away from the fault ends, indicating relatively

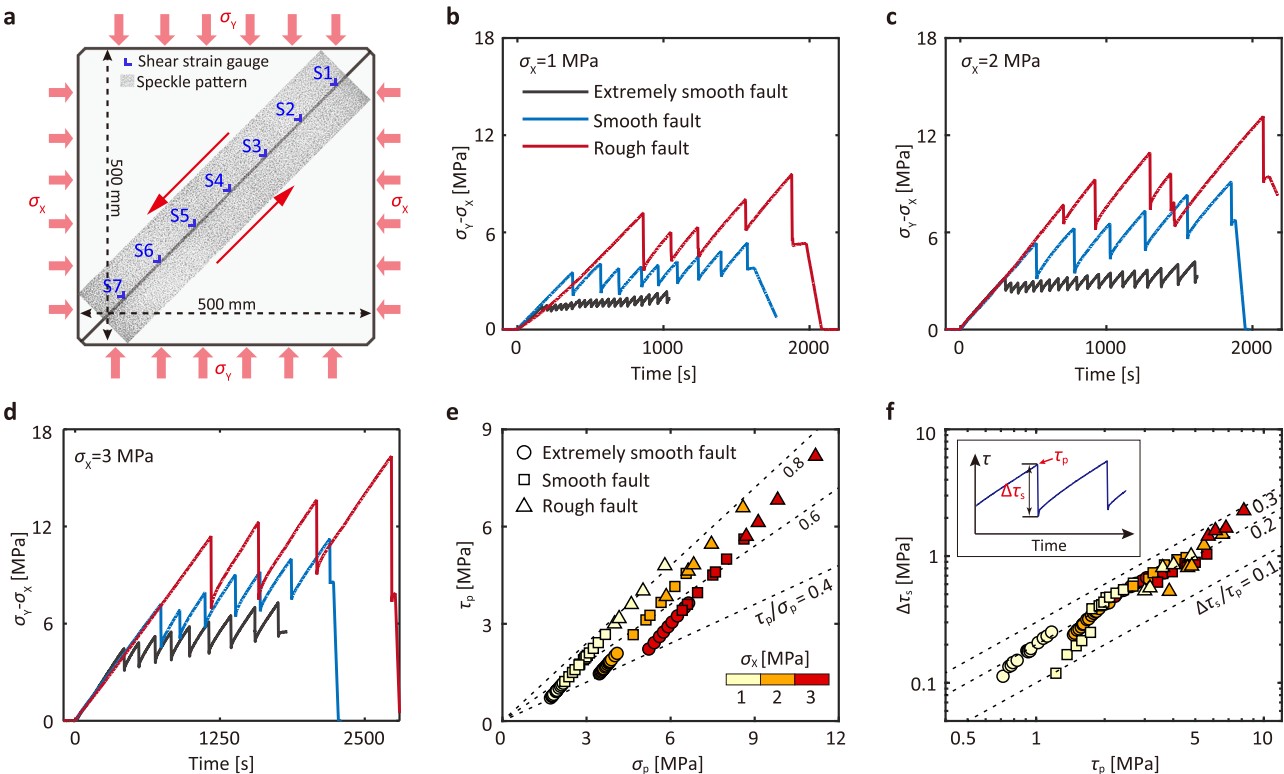

**Fig. 1 | Stick-slip events mimicking natural earthquakes. a** Schematic of laboratory earthquake model. The laboratory fault, with total length of 650 mm and observation window of 550 mm, is subjected to biaxial loading. The fault slip directions are indicated by red arrows. **b–d** Differential stress ($\sigma_Y - \sigma_X$) as a function of time at increasing horizontal stress ($\sigma_X = 1$, 2, and 3 MPa) on extremely smooth (black), smooth (blue), and rough (red) faults. **e** Stress state, global peak shear stress $\tau_p$ as a function of peak normal stress $\sigma_p$, at the onset of instability for events on extremely smooth (cycles), smooth (squares), and rough (triangles) faults. The color of the symbol is coded by the horizontal stress $\sigma_X$. Dashed lines correspond to different ratios of $\tau_p/\sigma_p$. **f** Static shear stress drop $\triangle\tau_s$ as a function of peak shear stress $\tau_p$ for each event. Dashed lines indicate different values of relative shear stress drop $\triangle\tau_s/\tau_p$. The inset illustrates the definition of static shear stress drop $\triangle\tau_s$ and peak normal stress $\tau_p$. The symbol follows that in (**e**).

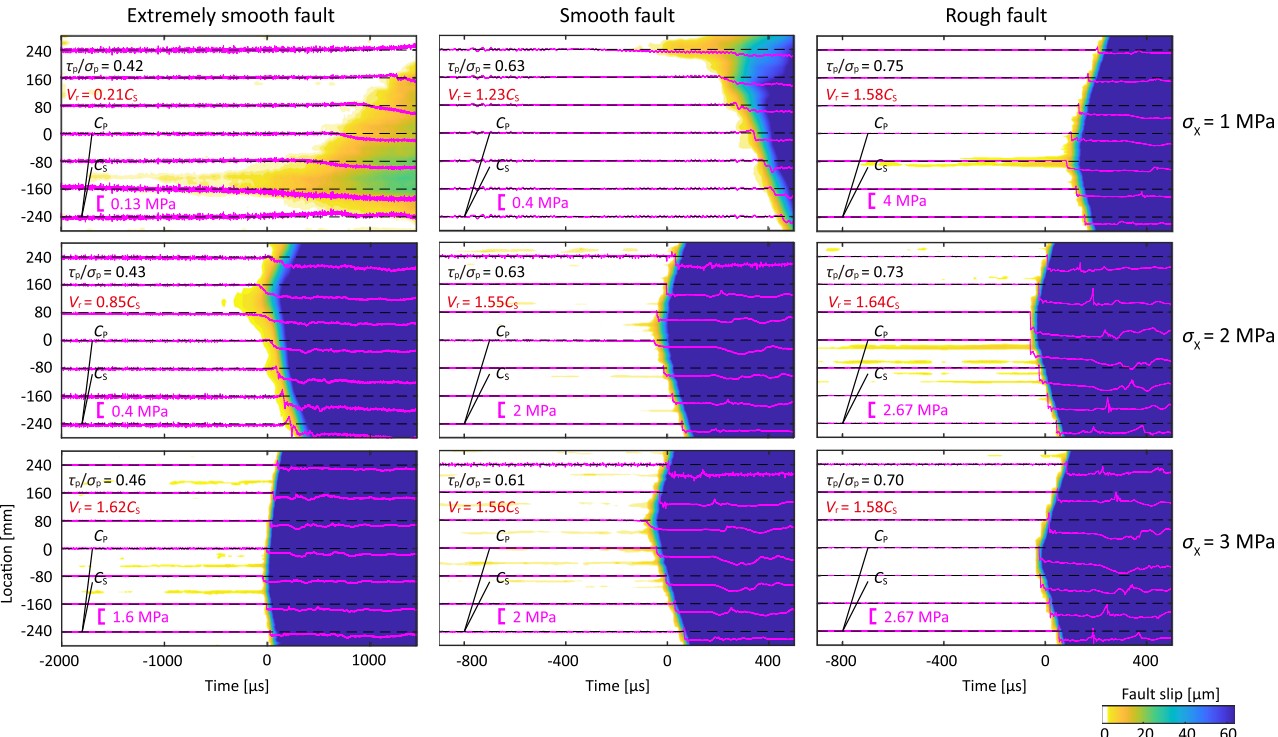

**Fig. 2 | Nucleation and rupture process of spontaneous events.** Each subgraph shows the space-time distribution of fault parallel slip. Colors indicate the distributed measurements of fault slip obtained from optical images. The magenta lines correspond to the shear stress measured using the shear strain gauges. The ratio of global peak shear stress to normal stress ($\tau_p/\sigma_p$) and the measured rupture speed $V_r$ normalized by S wave speed $C_S$ (see "Methods") of each event are indicated on each subgraph.

uniform distribution of the loading as further demonstrated in Supplementary Fig. 4, and the fault strength. The distribution of shear stress along the fault may be more uniform than that in previous direct shear experiments[26]. For events on the extremely smooth faults under low horizontal stress $\sigma_X$, the rupture initiates as a preslip zone expanding quasi-statically. This preslip zone is defined as the "nucleation zone" with its expansion representing the rupture nucleation phase. Dynamic rupture follows once the nucleation zone reaches a critical size. This observation is broadly consistent with the classic model of rupture nucleation, comprising three stages (prior to the unstable rupture), *viz*.: formation of the failure nucleus, quasi-static growth and acceleration[33,34]. With an increase in both the horizontal stress $\sigma_X$ and the fault roughness, the nucleation process becomes more complex, featuring more than one preslip zone, similar to numerical models with inhomogeneous static friction[35]. As preslip tends to develop where the local shear stress exceeds the frictional strength, it is plausible that fault roughness similarly may result in a slightly heterogeneous distribution of the fault strength, which is initially uniform. In addition, the critical size of the nucleation zone decreases with increasing $\sigma_X$. Thus, in these experiments, the nucleation zone on the rough fault with $\sigma_X = 3$ MPa is too small to be visible as limited by the spatial resolution of the measurement method.

Figure 2 also shows that the measured dynamic rupture speed (see "Methods" section) depends directly on both $\sigma_X$ and the fault roughness. Rupture on the extremely smooth faults at $\sigma_X = 1$ MPa propagates upward at a nearly constant velocity of ~300 m/s ($0.21C_S$). In contrast, ruptures on rough faults with higher $\sigma_X$ always attain supershear speed ($>C_S$). Regardless of the speed, these ruptures attain their final propagation speed immediately following spontaneous nucleation. No intermediate phase of acceleration preceding the dynamic rupture propagation is detected in these results. Furthermore, the supershear transition by the Burridge-Andrews mechanism, predicted by numerical simulations[21] and

confirmed by laboratory experiments with similar artificial abrupt nucleation[23], is absent. Our observations are also different from numerical results featuring direct supershear transition on sufficiently stressed faults[24], where the rupture speed increases continuously through the forbidden zone. Importantly, our experiments reproduce the early supershear rupture of the 2018 Palu earthquake. This spontaneous nucleation may be responsible for the absence of the speed transition proposed in the literature featuring numerical or experimental abrupt triggering. Such immediate transition to supershear has also been observed from some experiments[36–38], but with limited resolution. In this study we focus on one-dimensional homogenous faults subjected to uniform loading. For one-dimensional heterogeneous faults or two-dimensional homogeneous faults, other nucleation mechanisms for supershear ruptures are possible[39–41]. Furthermore, nearly steady subEshelby supershear speeds are observed (Supplementary Figs. 5 and 6), which confirms the other puzzle in the 2018 Palu earthquake. Although rupture speeds between $C_S$ and $C_E$ have been reported[38,42], the observed rupture speeds are always transient. These observations of non-steady rupture speed may be associated with the stress gradient along the fault in direct shear experiments[26] as first pointed out by Scholz[43,44]. If the rupture initiates from the edge of the direct shear specimen that suffers the most from the stress nonuniformity, then such an issue may persist even if the stress is uniform in the central part of the fault[45].

## Factors governing rupture speed

The ratio of shear stress to normal stress[26] and shear stress drop[29,46] have been proposed as the key factors that control the rupture speed. To explore the factors governing the rupture speed, a total of 81 stick-slip events (Supplementary Data 1) are summarized in Fig. 3, enabling verifications of existing theoretical conjectures. Following Ben-David et al.[26], we first examine the normalized rupture velocity relative to the

**a**

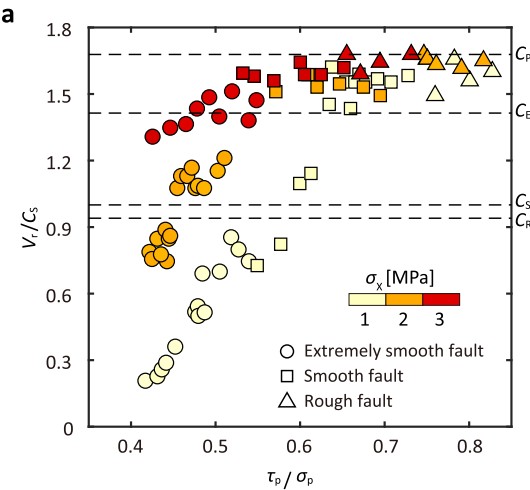

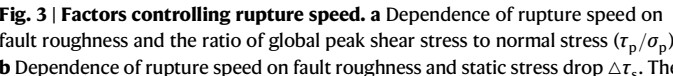

**b**

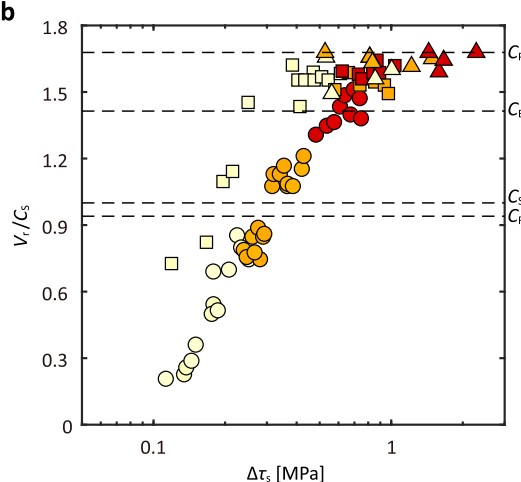

**Fig. 3 | Factors controlling rupture speed. a** Dependence of rupture speed on fault roughness and the ratio of global peak shear stress to normal stress ($\tau_p/\sigma_p$). **b** Dependence of rupture speed on fault roughness and static stress drop $\triangle\tau_s$. The rupture speed $V_r$ is normalized by S wave speed $C_S$. In (**a**) and (**b**), symbols represent rupture events on extremely smooth (circles), smooth (squares), and rough (triangles) faults, with color coded by horizontal stress $\sigma_X$.

ratio of $\tau_p/\sigma_p$. As shown in Fig. 3a, the rupture velocity increases with $\tau_p/\sigma_p$, which is in general consistent with their prior results. However, we observe that the specific relationship between $\tau_p/\sigma_p$ and $V_r$ also varies with $\sigma_X$. This phenomenon was either not evident or obscured within their results[26] where data were broadly scattered due to stress inhomogeneity. In our observations, dynamic rupture propagation speeds are steady (Fig. 2) and are absent from the theoretically predicted forbidden zone ($C_R < V_r < C_S$) (Fig. 3)—a feature not honored in existing results[26,29,30], possibly due to uncertainties in rupture speed measurements. Moreover, our results show that fault roughness plays an important role in determining dynamic rupture speed. Despite significant scatter, Passelègue et al.[29] proposed that the rupture speed may be related to the static stress drop following a similar argument based on field data[47]. Our results also show a consistent increase in the rupture speed with the static stress drop, but the relation between these two quantities is apparently modulated by the fault roughness (Fig. 3b).

Within the framework of linear elastic fracture mechanics, rupture propagation is governed by the energy balance[48]. According to this rule, a theoretical model has been derived for subRayleigh ruptures, where the rupture speed is correlated with $\triangle\tau_d^2/\sigma_N$, with $\triangle\tau_d$ and $\sigma_N$ being the dynamic stress drop and the normal stress (i.e. $\sigma_p$), respectively (see "Methods")[28]. This useful conclusion remains conjectural since they adopted a two-dimensional theory to model their three-dimensional triaxial tests. Here we examine our experimental data against their conjecture, with $\triangle\tau_d$ being quantified at high fidelity by the shear strain gauges near the fault (Fig. 1a, Supplementary Fig. 3 and inset of Fig. 4a). A consistent trend is obtained for all levels of fault roughness (Fig. 4a). Our experimental results not only verify this conjecture, but also extend it from the subRayleigh to both subRayleigh and supershear regions.

**Self-similar analysis**

Steady-state fracture models have been widely used to analyze dynamic rupture[12,49]. In this model, for fractures with a given speed, the mechanical fields are assumed to remain invariant with respect to the reference-frame of the moving crack-tip. Because the mechanical fields (including the stress field) are invariant, the stress intensity factor remains the same. Constant stress intensity factor and constant fracture velocity result in constant fracture energy in this model. Cracks propagating in speed regimes $V_r < C_R$ and $C_S < V_r < \sqrt{2}C_S$ are shown to be unstable, while the speed regime $\sqrt{2}C_S < V_r < C_P$ is stable[12,13]. More generally, stable rupture speed is not accessible for simple

models with constant fracture energy[20,50]. Therefore, observations of ruptures growing bilaterally at roughly constant speeds (Supplementary Figs. 5 and 6) over a wider range of rupture speeds imply that the steady-state model cannot interpret our experiments. Furthermore, as the nucleation zone size is negligible in comparison to the fault length, and our one-dimensional homogenous laboratory faults are subjected to uniform loading, we can confidently treat our experiment as a mode II self-similar crack problem[51,52], which has been partially adopted to analyze supershear ruptures[42].

As detailed in "Methods", the theoretical analysis is valid for both self-similar subRayleigh and supershear ruptures. The specific fracture energy $\Gamma_{II}$ can include not only the fracture energy at the rupture tip, but also other tip-localized dissipation[53]. Experiments have revealed that specific fracture energy may be a function of the rupture speed[54] and rupture length[55], at least for mode I ruptures. We assume that the specific fracture energy is equivalent to the fracture energy of a rupture that obeys the classical slip-weakening law[21]. By further using the relation between the critical slip distance $D_c$ and the rupture length by Passelègue, et al.[28], we arrive at a unique relation between $\triangle\tau_d^2/\sigma_N$ and $\beta$, with the former being the generalized driving load and the latter being the rupture speed normalized by the P wave speed. This theoretical result nicely confirms the experimental dependence of the rupture speed on the driving load (Fig. 4a). We further fit the experimental data using the theoretical model (Fig. 4a), where the consistency between the model prediction and experimental data suggests that the laboratory earthquake ruptures are indeed self-similar.

These same data (Fig. 4a) show stable rupture speeds from ~0.2$C_S$ to near $C_R$ and from $C_U$ (~1.08$C_S$) up to $C_P$, consistent with field observations[20]. However, such stable speed regions are not compatible with those predicted by the steady-state rupture model with constant fracture energy[12], where only $C_R$ in the subRayleigh region and supershear speeds above $C_E$ are deemed stable. A rupture is stable when an increase in rupture speed requires an increase in the driving load ($\triangle\tau_d^2/\sigma_N$ in the current case)[12]. Following this rule, we can divide the rupture speed spectrum into several ranges (Fig. 4a). Ruptures with speeds falling in the ranges of $V_r < C_R$ and $C_U < V_r < C_P$ are all stable, while ruptures with speeds in the range $C_S < V_r < C_U$ are unstable. Our results confirm that even a simple homogeneous fault can allow a spontaneous rupture to propagate stably in the conventionally deemed unstable regions of $C_S < V_r < C_E$ and $V_r < C_R$. Unlike previous attempts to reconcile the contradiction between conventional theory and field observation by invoking geometrically sophisticated fault models[16,18,20], we provide an elegant and simple resolution here.

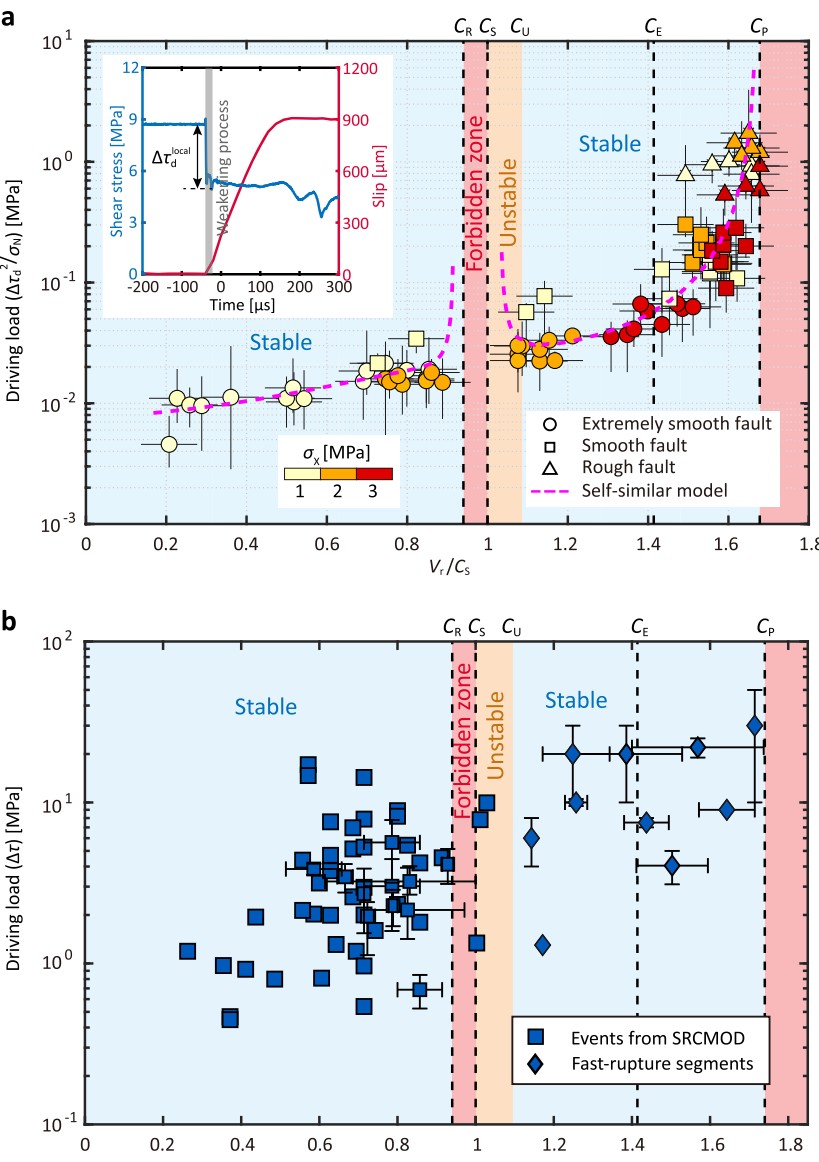

**Fig. 4 | Rupture speed versus driving load. a** Normalized rupture speed $V_r/C_S$ of laboratory earthquakes as a function of the driving load $\triangle\tau_d^2/\sigma_N$ with speed stability regions revised. The inset shows the definition of the local dynamic stress drop $\triangle\tau_d^{local}$, with the gray strip delineating the transient weakening process. Rupture events on extremely smooth, smooth, and rough faults are represented by circles, squares, and triangles, respectively. The color of the symbol is coded by the horizontal stress $\sigma_X$. Horizontal bar indicates the range of rupture velocity. Vertical bar indicates the range of driving load calculated from not perfectly uniform local dynamic stress drop $\triangle\tau_d^{local}$. The dashed magenta lines represent model predictions based on the self-similar fracture analysis. **b** Normalized rupture speed $V_r/C_S$ of compiled natural strike-slip earthquakes as a function of the driving load represented by stress drop $\triangle\tau$. Strike-slip events (squares) are chosen from the SRCMOD database[43] (Supplementary Data 2) with the fast-rupture segments (diamonds) compiled from well-instrumented events (Supplementary Data 3). Error bars indicate the uncertainties from the references. Details in estimating the stress drop are given in the supplemental materials. Despite their scatter, the field data match well with the speed regions delineated by the theoretical model, that is in turn congruent with the laboratory data.

## Discussion

Mature faults with large slip offsets are typically smoother than short-offset immature faults[56]. Congruently, high rupture speed is correlated with mature fault sections[57]. This is qualitatively consistent with previous experimental observations that increasing fault roughness has a stabilizing effect on fault rupture[58] - seemingly contrary to our findings, here. This discrepancy may arise from the limited range of roughness involved in this study. Recently, the effect of fault roughness on rupture dynamics has been resolved over a wide range of roughnesses[59]. This has defined a critical roughness, below which the stress drop increases with roughness, and beyond which the stress drop decreases with roughness. This suggests that an increase in roughness may not necessarily stabilize fault rupture. In this work, the

fault is overall smooth (Supplementary Fig. 1). Thus, an increase in fault roughness may promote fault instability according to observations by Morad, et al.[59]. Our results are generally consistent with those of Xu et al.[31], who demonstrated that rough faults could indeed accommodate supershear rupture.

Finally, we compare our model with observations of natural earthquakes. Generally, strike-slip earthquakes on elongated faults may be simplified as mode-II shear ruptures−replicated by, and most relevant to, our laboratory geometry. Since determining the absolute normal stress for the source zone of natural earthquakes is intractable, the driving load $\triangle\tau_d^2/\sigma_N$ is replaced by the stress drop $\triangle\tau$ for natural earthquakes. Note that dynamic stress drop of laboratory earthquakes may be taken as equivalent to "seismological" stress drop of natural

earthquakes (Methods). A compilation of the average rupture velocities and estimated stress drop of strike-slip earthquakes is available from the SRCMOD database[60] (Supplementary Data 2), together with fast ruptures on fault segments of some well-instrumented earthquakes (Supplementary Data 3). As seen in Fig. 4b, most earthquakes assume average rupture speeds slower than the Rayleigh wave speed. However, ruptures on certain fault segments can propagate at speeds faster than the S wave speed of the crust (~3.5 km/s). Supershear ruptures with speeds less than the Eshelby speed are indeed accessible for natural earthquakes. The speeds of natural earthquakes fall nicely into the stable speed ranges delineated by the self-similar earthquake model. While there are considerable uncertainties in the resolved source parameters, the rupture speed of natural earthquakes is positively correlated with the stress drop, which is generally consistent with our experimental results. We also note that some observations from natural earthquakes indicate that rupture speed may decrease with an increase in the stress drop, especially for earthquakes with subRayleigh rupture speeds[61,62]. However, the validity of this finding may suffer from tradeoffs and uncertainties in estimating source properties for natural earthquakes.

We use highly instrumented laboratory experiments to constrain magnitudes and mechanisms governing rupture speeds on one-dimensional laboratory faults with homogeneous strength and uniform tectonic loading. Our experimental results reveal an abrupt jump to supershear rupture immediately following a spontaneous nucleation stage, countering expectations of the Burridge-Andrews transition by a speed jump over the forbidden speed zone, or the direct transition through that zone by continuous acceleration. Secondly, contrary to prior characterizations, we observe that ruptures can propagate stably over nearly the full range of subEshelby speeds and that the rupture speed is directly conditioned by the magnitude of the driving load ($\triangle\tau_d^2/\sigma_N$). These dynamic observations conform to a self-similar model of dynamic rupture that spans the full range of rupture speeds, which neatly explains the stable slow-supershear rupture of the 2018 $M_w$ 7.5 Palu earthquake and stable subRayleigh ruptures for other natural earthquakes. Thus, this model is of fundamental importance in the interpretation of the spectrum of earthquake rupture speeds, in defining critical controls of rupture speeds, in revealing nucleation mechanisms, and therefore in the assessment of seismic hazards. Because natural faults are more complex than laboratory faults, alternative models in the framework of linear elastic fracture mechanics coupled with velocity-dependent fracture energy[63] or non-constant fracture energy[53] may also work.

# Methods

## Laboratory faults

Polymethyl methacrylate (PMMA) has been commonly used as an analog to rock material in laboratory earthquake investigations[26,64,65]. In this work, square PMMA plates with a dimension of 500 mm × 500 mm × 20 mm is used. PMMA has the physical and mechanical properties as follows: density $\rho = 1190$ kg/m$^3$, the Young's modulus $E = 6.24$ GPa, the shear modulus $\mu = 2.40$ GPa, and the Poisson's ratio $v = 0.3$. The seismic wave velocities are $C_S = 1.43$ km/s and $C_P = 2.40$ km/s for plane stress conditions.

The square PMMA plate is cut diagonally into two identical triangular plates, using a computer-numerical control (CNC) engraving and milling machine. Then the cut faces are polished to remove the machining lines. Subsequently, the polished surfaces are roughened by sandblasting using white fused alumina particles for a specified duration, pressure, and blasting distance. Three kinds of grain size are utilized in blasting, resulting in planes with three kinds of roughness as quantified by a stylus surface profilometer. The measured root mean square (RMS) heights are ~0.1 μm, ~1 μm, and ~5 μm, respectively (Supplementary Fig. 1a). We obtain the power spectral density (PSD) of the roughness by performing fast Fourier transform (Supplementary

Fig. 1b). For a self-affine surface, the PSD value $p$ follows a power law with the wavelength $\lambda$

$$p = C\lambda^\alpha \tag{1}$$

where $C$ and $\alpha$ are constants. The value of $\alpha$ is 1.15, 1.87, 2.62 for the extremely smooth, smooth, and rough fault, respectively. It can be seen that a rougher fault has a higher $\alpha$ value, consistent with existing studies[56].

## Experimental setup and testing procedure

Four independently controlled hydraulic rams are used to apply simulated tectonic loads to the fault model (Supplementary Fig. 2). First, the horizontal stress ($\sigma_X$) and the vertical stress ($\sigma_Y$) are increased to a specified stress level at the same rate of 0.01 MPa/s. After a hold of 15 minutes, $\sigma_X$ is held constant at the specified stress level while the vertical rams are switched to displacement-controlled mode at a rate of 1 μm/s. With the movement of vertical rams, $\sigma_Y$ increases quasi-statically and so does the differential stress ($\sigma_Y$-$\sigma_X$). This biaxial loading configuration can achieve a relatively uniform normal stress and shear stress along the fault, reducing the influence of stress gradients on the nucleation process and subsequent rupture behaviors[43]. During experiments, both $\sigma_X$ and $\sigma_Y$ are controlled and measured by load cells to a resolution of 0.005 MPa. The displacement of the rams is controlled and measured with a resolution of 1 μm. Both stress and displacement are recorded at a sampling rate of 100 Hz.

## Diagnostic methods

We use a high-speed camera (Photron FASTCAM SA1.1) to capture the image of the speckle pattern on the front side of the specimen during rupture nucleation and propagation (Supplementary Figs. 2 and 3a). Two LED lamps provide stable high intensity illumination for imaging. The camera is triggered by the acoustic emission signal by a piezoelectric acoustic sensor mounted on the specimen. The framing speed of the camera is set as 50,000 frames/s and the exposure time as 2 μs. The digital images recorded have a spatial resolution of 1024 × 112 pixels. The field of view is 600 mm in length and thus one pixel represents a physical dimension of about 600 μm. The digital image correlation (DIC) method is used to reveal the spatiotemporal evolution of the fault-parallel slip during the rupture (Supplementary Fig. 3b). This technique has been successfully applied in recent laboratory earthquake experiments[66,67]. Following an error analysis of DIC[68], we determine that the full-field displacement measurement has a standard deviation of 2 μm.

In addition to the full-field displacement measurement using the DIC method, seven shear strain gauges are mounted at 8 cm separation on the backside of the specimen along the fault to record the shear strain (Supplementary Fig. 3c). The strain gauge is 1 mm away from the linear fault. The strain signals are amplified by an amplifier (gain ~200, bandwidth ~1 MHz) and sampled at a rate of 1 MHz by a digital acquisition system. The digital acquisition is also triggered by the acoustic emission signal. Therefore, the strain signals are synchronized with the digital images. A schematic of the shear strain gauge is shown in Supplementary Fig. 3d. The shear strain gauge has two components, one component is in the vertical direction and the other is in the horizontal direction. Denoting the measured strain components as $\varepsilon_1$ and $\varepsilon_2$, the shear strain ($\varepsilon_{XY}$) along the fault is

$$\varepsilon_{XY} = 1/2(\varepsilon_1 - \varepsilon_2) \tag{2}$$

## Determination of rupture velocity

Taking advantage of the simultaneous measurements of the full-field displacement and local shear strains, we can determine the rupture speeds by two independent methods. First, the full-field DIC

displacement measurement captures the slip distribution along the fault at different instants in time. The rupture front is identified as the point where the fault-parallel particle velocity $\dot{u}$ exceeds a threshold $\dot{u}_{th}$. We first calculate the full-field fault-parallel particle velocity $\dot{u}$ by temporal differentiation at points along the fault, at an interval of ~0.6 mm (=1 pixels) (Supplementary Fig. 5). Since the error in displacement is ~2 μm, the error in particle velocity is ~0.1 m/s (= 2 μm / 20 μs). We set $\dot{u}_{th} = 0.2$ m/s as the threshold for the passage of rupture front, allowing the arrival time of the rupture to be determined. Because the slip velocity increases sharply as the rupture passes, the arrival time is insensitive to the small change in $\dot{u}_{th}$. Finally, we can obtain the average rupture velocity within 20 μs by temporal differentiation. Thus, the error in rupture velocity is less than 30 m/s.

Second, from the local shear strain measurement at seven locations along the fault, the arrival time of the rupture front can be picked from the shear strain record with an error of only 1 μs. The rupture velocity between neighboring strain gauge arrays is computed as the interval (80 mm) divided by the difference in arrival time. For supershear ruptures propagating at velocities of ~2 km/s, the difference in arrival time is 40 μs. Thus, the relative error in rupture velocity is 2.5% (~50 m/s); for subRayleigh ruptures propagating at a velocity of ~1 km/s, the error is smaller (~25 m/s).

The along-fault rupture velocity calculated from these two independent methods is shown in Supplementary Figs. 5 and 6. These independently evaluated velocities match closely and with nearly constant velocity - as there is no sharp accelerations or decelerations. The variation in rupture velocity is small and usually within ~0.1 km/s— likely resulting from small variations in applied fault stress (Supplementary Fig. 4). The average rupture velocity is computed by averaging the local rupture velocity along strike.

## Static and dynamic stress drop

In this work, static stress drop $\triangle\tau_s$ is defined as the global drop in the macroscopic shear stress resolved on the fault (inset in Fig. 1f) - similar to previous studies[69]. However, static stress drop may not accurately quantify the change in shear stress associated with the rupture. This is because static stress drop includes the contributions from both the specimen and the load frame "machine stiffness" during unloading[37,59]. Furthermore, static stress drop is an integration of stress change resulting from multiple ruptures during a stick-slip event. The free boundary at the end of the fault is subject to surface breakout when a rupture arrives[31], potentially allowing backward-propagating re-ruptures, as shown in Supplementary Fig. 5. Thus, the globally measured static stress drop $\triangle\tau_s$ cannot represent the stress drop accompanying the first rupture during a stick-slip event. To isolate the stress change of the first rupture from the global value, we measure the local dynamic stress drop $\triangle\tau_d^{\text{local}}$ of the first rupture from the evolution of the shear stress, as shown in the inset in Fig. 4a. This treatment is in accordance with previous studies[30,37]. The dynamic stress drop $\triangle\tau_d$ is achieved by averaging the local dynamic stress drop along strike.

For natural earthquakes, because the faults are embedded in a nearly infinite medium (the crust), boundary-induced repeated ruptures are likely absent. Thus, the dynamic stress drop $\triangle\tau_d$ of laboratory earthquakes is equivalent to the "seismological" stress drop $\triangle\tau$ in the investigation of natural earthquakes, especially for ruptures on simple faults.

## Self-similar model for subRayleigh and supershear ruptures

Rupture dynamics is governed by energy balance. For mode II shear rupture with constant specific fracture energy (i.e., total energy dissipation), the rupture will accelerate towards the limiting speed $C_R$, and some of them could suddenly jump into supershear regime with the final speed approaching the other limiting speed $C_P$[20]. However, a constant rupture speed has been observed in seismological observations[17]. This constant rupture speed requires an increase in dissipated energy with growing rupture length[53]. Several mechanisms have been proposed to provide this increase, including off-fault yielding[53], provision of specific friction laws[70,71] and slip-rate-dependent fracture energy[63]. Experiments have revealed that for constant rupture speed, the dissipated energy increases with rupture length, at least for mode I ruptures[55].

In a self-similar rupture model, an inherent theoretical consequence is that the specific fracture energy $\Gamma_{\text{II}}$ must increase with the rupture length[72]. $\Gamma_{\text{II}}$ is a generalized concept that may contain energy consumed at the rupture tip (fracture energy) and off-fault inelastic dissipation[53]. The proportionality between $\Gamma_{\text{II}}$ and rupture length is consistent with dynamic rupture models with off-fault dissipation[50,53,72]. Also, this proportionality is consistent with field observations that "seismological" fracture energy (breakdown work) scales with the rupture size[73].

According to the theory of dynamic fracture mechanics[52], the energy release rate for a mode II self-similar rupture is

$$G_{\text{II}}(a, \triangle\tau_d, \beta) = G_{\text{II}}^s(a, \triangle\tau_d, 0)g_{\text{II}}(\beta) \qquad (3)$$

where $\beta = V_r/C_P$ is the normalized rupture speed, $g_{\text{II}}(\beta)$ is a universal function that describes the effect of rupture speed on the energy release, $G_{\text{II}}^s(a, \triangle\tau_d, 0) = \frac{\pi(\triangle\tau_d)^2 a}{4(1-k^2)\mu}$ is the energy release rate for a static mode II rupture with $\triangle\tau_d$ being the dynamic stress drop, $k$ the ratio between $C_S$ and $C_P$ for plane stress conditions, $a$ the crack half length, and $\mu$ the shear modulus.

For self-similar rupture, the specific fracture energy $\Gamma_{\text{II}}$ increases with the rupture length[72]. Another essential feature of specific fracture energy is its dependency on rupture velocity, which has been verified in experiments, at least for mode I ruptures[54,74]. Such dependency is also implied in the cell model[72]. Taking the above features into account, we obtain a generic expression for the specific fracture energy

$$\Gamma_{\text{II}} \propto af(\beta) \qquad (4)$$

where $f(\beta)$ is an unknown function that describes the effect of rupture speed on the fracture energy.

We further assume that specific fracture energy is equivalent to energy dissipation for a rupture that obeys the classical slip-weakening law, enabling $\Gamma_{\text{II}}$ to be written as

$$\Gamma_{\text{II}} = \frac{1}{2}\triangle\tau_b D_c \qquad (5)$$

where $D_c$ is the critical slip distance, $\triangle\tau_b$ ($= \tau_p - \tau_r$) the breakdown stress drop[75], $\tau_p$ denotes the peak shear stress and $\tau_r$ the residual shear stress in the slip-weakening model. Further, $\triangle\tau_b = \tau_p - \tau_r = \sigma_N(f_p - f_r)$, where $\sigma_N$ is the normal stress(i.e. $\sigma_p$ in our experiments), and $f_p$ and $f_r$ the peak and the residual friction coefficients, respectively.

A slip-weakening model with scale-dependent $D_c = \eta a$ has been utilized for analyzing earthquake ruptures[53,70]; such assumption can be considered as a lumped representation of the fracture energy, off-fault dissipation, and non-edge-localized dissipation. Thus, Eq. (5) can be expressed as

$$\Gamma_{\text{II}} = \frac{1}{2}\sigma_N(f_p - f_r)\eta a \qquad (6)$$

where $\eta$ is a dimensionless parameter that may be dependent on the rupture speed (section 4.5 in reference 53). So far, the methodology is similar to that used to model laboratory earthquake experiments in triaxial tests[28].

Comparing Eqs. (4) and (6), $\Gamma_{II}$ can be rewritten as

$$\Gamma_{II} = \sigma_N f(\beta) a \tag{7}$$

where $f(\beta) = \frac{1}{2}\left(f_p - f_r\right)\eta$ is introduced to jointly consider the effect of rupture speed on the breakdown friction drop and $\eta$. Combining Eqs. (3) and (7), the energy balance $G_{II} = \Gamma_{II}$ leads to

$$f(\beta) = \frac{\pi}{4\left(1-k^2\right)\mu} g_{II}(\beta)\frac{\triangle\tau_d{}^2}{\sigma_N} \tag{8}$$

For a subRayleigh rupture, the analytical expression for $g_{II}(\beta)$ is (equation 6.9.92 in reference 52)

$$g_{II}(\beta) = \frac{2\left(1-k^2\right)\sqrt{\left(k^2-\beta^2\right)^3}R(\beta)}{k\beta^2\left[g_2(\beta)\right]^2} \tag{9}$$

where $k = C_S/C_P$, $R(\beta) = 4k^3\sqrt{1-\beta^2}\sqrt{k^2-\beta^2}-\left(2k^2-\beta^2\right)^2$ is the Rayleigh function, and $g_2(\beta) = \beta^{-2}\left[8k^2\left(k^2-\beta^2\right)\right]\boldsymbol{E}\left(\sqrt{1-\beta^2}\right) - 4k^2\left(k^2-\beta^2\right)\boldsymbol{K}\left(\sqrt{1-\beta^2}\right) - \beta^{-2}\left[8k^4-8k^2\beta^2+\beta^4\right]\boldsymbol{E}\left(\sqrt{1-\beta^2/k^2}\right) + \left(4k^2-3\beta^2\right)\boldsymbol{K}\left(\sqrt{1-\beta^2/k^2}\right)$, in which $\boldsymbol{K}$ and $\boldsymbol{E}$ are complete elliptic integrals of the first and second kind.

The values of $f(\beta)$ as a function of $\beta$, evaluated from our experimental results, are shown in Supplementary Fig. 7. In particular, $f(\beta)$ approaches zero as $\beta$ is close to 0.56 (corresponding to $C_R$). We fit the experimental results by the following function in the subRayleigh region:

$$f_1(\beta) = 3.925\left(1-\left(\frac{\beta}{C_R/C_P}\right)^{8.129}\right)\times 10^{-6} \tag{10}$$

This equation automatically ensures that $f_1(C_R/C_P)$ is zero. Combining Eqs. (5) and (6), we can evaluate the relationship between $\triangle\tau_d{}^2/\sigma_N$ and $V_r$. This function matches well our experimental data at subRayleigh speeds (Fig. 4a).

For a supershear rupture, the analytical expression for the energy release rate is more involved. In the framework of linear elastic fracture mechanics, there would be no energy flux into the crack tip for mode II cracks propagating at an intersonic (supershear) velocity, except when $V_r = C_E$, the Eshelby speed. By introducing a Barenblatt-type cohesive zone, the energy flux into the crack tip becomes positive and all supershear rupture speeds become physically acceptable. According to Broberg[52,76], $g_{II}(\beta)$ for a supershear self-similar rupture with a cohesive zone at the crack tip (equation 6.9.124 in reference 52 and equation 68 in reference 76) is

$$g_{II}(\beta) \approx 4\left(1-k^2\right)B(\beta)\Gamma_D(\gamma)\left(\frac{d}{a}\right)^{1-2\gamma} \tag{11}$$

where d is the size of the cohesive zone, $B(\beta)$ and $\Gamma_D(\gamma)$ are known functions: $B(\beta) = \frac{\beta^2 F_1(\beta)e^{-2I_0(1/\beta)}}{2^{2-2\gamma}}\frac{\sin\pi\gamma}{\gamma^2 N^2}\left(\frac{1-\beta}{1+\beta}\right)^{2\gamma}$, $\Gamma_D(\gamma) = 2\left[\int_0^1\frac{D(-u)}{u^{1-\gamma}}du\right]^{-2}w_D(\gamma)$, with

$$\gamma(s) = \frac{1}{\pi}\tan^{-1}\frac{4k^3\sqrt{1-1/s^2}\sqrt{1/s^2-k^2}}{\left(1/s^2-2k^2\right)^2}$$

$$I_0(s) = p.v. 2\int_1^{1/\beta}\frac{\xi\gamma(\xi)-1/\beta\gamma}{\xi^2-s^2}d\xi$$

$$F_1 = \frac{\beta^2\sin\pi\gamma}{4k^2\sqrt{1-\beta^2}}$$

$$N = \int_1^{\frac{1}{\beta}}\frac{M\left(\frac{1}{\beta}\right)\left(\frac{2/\beta}{1/\beta-s}\right)^\gamma - M(s)\left(\frac{1/\beta+s}{1/\beta-s}\right)^{\gamma/\beta s}}{1/\beta-s}ds + \frac{M\left(\frac{1}{\beta}\right)\left(\frac{2}{\beta}\right)^\gamma}{\gamma(1/\beta-1)^\gamma}$$

$$M(s) = \frac{\sin[\pi\gamma]e^{-2I_0(s)}}{1/\beta+s}\left(\frac{s-1}{s+1}\right)^{\gamma/\beta s}$$

$$w_D(\gamma) = \int_0^1\xi^{1-\gamma}D(-\xi)\left\{\int_0^1\frac{D(-\xi)-D(-u)}{u^{1-\gamma}(u-\xi)}du + \int_1^\infty\frac{D(-\xi)du}{u^{1-\gamma}(u-\xi)}\right\}d\xi$$

$$D(-\xi) = 1-\xi$$

$\gamma = \gamma(1/\beta)$ is implied when the argument is omitted.

We estimated the ratio between $d$ and $a$ based on the shear stress change during the dynamic rupture process, as shown in Supplementary Fig. 8. Generally, $d/a$ falls in the range between 0.03 and 0.05. As shown in Supplementary Fig. 9, a small change in $d/a$ does not significantly change the normalized energy release rate[76]. Therefore, we assume that the ratio between $d$ and $a$ is a constant ($\approx 0.04$).

Combining Eqs. (8) and (11), we recover a relationship between $\beta$ and $f(\beta)$ for supershear ruptures, which is illustrated in the Supplementary Fig. 7. We fit the data by the following relation:

$$f_2(\beta) = \frac{0.0009\left[\left(\frac{\beta}{C_S/C_P}\right)^{0.0013}-1\right]}{1-\beta^{14.42}} \tag{12}$$

Note that the boundary condition that $f_2(C_S/C_P)$ is equal to zero is enforced by this equation. After substituting $f_2(\beta)$ into Eq. (8), we evaluate the relationship between $\triangle\tau_d{}^2/\sigma_N$ and $V_r$ numerically. The result is shown as the magenta dashed line in the supershear region in Fig. 4a.

## Estimates of rupture speed and stress drop for natural earthquakes

The data for natural earthquakes are mainly adopted from the SRCMOD database of finite-fault rupture models[60]. SRCMOD is an open-source database including more than 300 finite fault rupture models. For the models included in SRCMOD, we only consider the strike-slip events that can be approximately regarded as mode II ruptures (Supplementary Data 2). Except for the stress drops ($\Delta\tau$), key source parameters, including the geometry of the fault, the seismic moment ($M_0$), and the rupture velocity ($V_r$) are provided in the finite fault rupture models. The analytical relation between the fault rupture length ($L_r$), fault rupture width ($W_r$) and the seismic moment ($M_0$)[77] allows one to estimate the stress drop ($\Delta\tau$) and to explore the correlation between $V_r$ and $\Delta\tau$. For a circular fault with radius $r$ embedded in homogenous material, the analytical solution of the stress drop is

$$\triangle\tau = 7M_0/16r^3 \tag{13}$$

Whereas for a rectangular fault with large aspect ratio ($L_r \gg W_r$), the stress drop becomes

$$\triangle\tau = 2M_0/\pi L_r W_r^2 \qquad (14)$$

However, the exact application range of (14) is ambiguous. Recently, Luo et al.[78] proposed a more unified model that considers the transition of scaling relation between seismic moment ($M_0$) and rupture area ($A$). According to this model, we can estimate the stress drop from $M_0$ and rupture dimension ($L_r$ and $W_r$) as

$$\triangle\tau = \begin{cases} C\left(\frac{L_r}{W_r}\right)\frac{M_0}{A^{3/2}} & L_r \leq W_r \\ C\left(\frac{L_r}{W_r}\right)\frac{M_0}{AW_r} & L_r > W_r \end{cases} \qquad (15)$$

where $C\left(\frac{L_r}{W_r}\right) = \frac{\pi}{8}\left(7 - \frac{7-16/\pi^2}{1+(2W_r/L_r)^2}\right)$.

Noting that the rupture dimension in SRCMOD is always larger than the real rupture area on the fault. This oversized rupture dimension leads to an underestimate of stress drop when using Eq. (15). One way to improve the estimation is to trim the finite fault model to better approximate the true source dimension. Several methods have been proposed to determine the effective rupture area[79,80]. Here, we use a simple method to obtain the effective rupture area. In this method, the sub-faults with cumulative slip less than 5% of the maximum fault slip are eliminated. In addition, we assume that the aspect ratio of the finite fault model is representative of the real one. Therefore, combining the area of the trimmed fault and the original fault aspect ratio enables the rupture length ($L_r$) and rupture width ($W_r$) to be determined. Then, following Eq. (15), we can estimate stress drop for the compiled strike slip earthquakes in SRCMOD. The results for the estimated stress drop combined with other source parameters are listed in Supplementary Data 2.

Furthermore, we compile data for several well-instrumented earthquakes (Supplementary Data 3) that featured supershear ruptures. For these events, we are concerned with the rupture speed and the stress drop on the fast-rupture segments. These data are derived directly from the rupture models of these earthquakes as reported in the literature.

## Data availability
Raw data for Figs. 1, 3 and 4 are provided in the Supplementary Data 1-3. The source data for all other figures and the original laboratory data have been deposited in Figshare under accession code (https://doi.org/10.6084/m9.figshare.18865673.v2). The natural earthquakes data is available at SRCMOD database (http://equake-rc.info/srcmod/).

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

## Acknowledgements
This study was supported by National Natural Science Foundation of China (NSFC) grants no. 42174061and 42141010 (K.X.), and Project funded by China Postdoctoral Science Foundation grant no. 2022TQ0319 (P.D.), and the Natural Sciences and Engineering Research Council of Canada (NSERC) through the Discovery grant no. 72031326 (K.X.). We thank Prof. Hiroo Kanamori from Caltech, Prof. Shamita Das from Oxford University, Prof. Emily Brodsky from UC Santa Cruz, and Dr. Yingjun Deng from Tianjin University for discussions and comments. We thank Shiqing Xu, Allan Rubin, the other anonymous reviewer, and the Editor for their constructive comments.

## Author contributions
K.X. supervised the study. P.D. performed the experiments. P.D. & Y.X. carried out the analysis. P.D. wrote the paper. K.X. edited it. K.X., P.D., D.E. and J.-P.A. contributed to the revision of the paper. All authors contributed to the data interpretation.

## Competing interests
The authors declare no competing interests.

## Additional information

**Peer review information** : *Nature Communications* thanks Shiqing Xu, Allan Rubin and the other, anonymous, reviewer(s) for their contribution to the peer review of this work. A peer review file is available.

