## [Peer Review File · Nature Communications]

Laboratory earthquakes decipher control and stability of rupture speedsREVIEWER COMMENTS

Reviewer #1 (Remarks to the Author):

By simulating spontaneous dynamic ruptures along a polymethyl methacrylate (PMMA) laboratory fault, Dong et al. provide new insights into the stability of rupture speed in both the sub-Rayleigh and supershear regimes. On the basis of self-similar solutions, they further identify a universal “driving force” that can fit and explain their experimental observations. Finally, the authors link their results to natural observations, showing that their analysis can also explain the rupture speed and source scaling relation for a suite of natural earthquakes along strike-slip faults. I find this manuscript interesting and suitable for publication in Nature Communications. It represents an excellent piece of work that combines experiments, theory and natural observations. I also think the results will attract a wide range of readers across geoscience, material science, natural hazard, etc.

In the meantime, I do have some concerns, mostly on the use of PMMA and the comparison to several previous studies. I don't think these concerns will affect the overall analysis or conclusion of this work, but just feel that some clarification and discussion are needed to reinforce the reasoning. After these concerns are properly addressed, I would be happy to make my further recommendation.

Major comments:

#1 The use of PMMA

(1.a) Previous studies show that PMMA has a viscoelastic behavior, whose effective elastic moduli (or wave velocities) could increase with the local strain rate (Gori et al., NatureComm, 2018). Since the exact values of wave velocities are important for defining the stability regimes of dynamic ruptures, I was wondering whether the authors have performed an accurate measurement of wave velocities for PMMA in the dynamic range, as done by Shlomai et al. (PNAS, 2020).

(1.b) Plane stress or plane strain

I may have missed it, but I find the information of plane stress or plane strain is never clearly mentioned in the text. This information not only affects the effective P wave velocity for a fault with finite width, but also influences the value of parameter “k” in the “Self-similar model for sub-Rayleigh and supershear ruptures” section in Methods. I therefore suggest adding the assumed condition for plane stress or plane strain explicitly.

(1.c) Transient versus self-similar rupture behaviors

Jay Fineberg's group has studied extensively the behaviors of laboratory earthquakes along a PMMA fault. Their results generally show a monotonically increasing trend of rupture speed towards some limit value, for both the sub-Rayleigh (Svetlizky et al., PRL, 2017) and the supershear regimes (Kammer et al., ScienceAdv, 2018; Svetlizky et al., JMPS, 2020). Although the asymptotic rupture speed in their studies does approach a constant value in each regime, the overall rupture behavior does not follow the strict definition for steady-state solution or self-similar solution, because the amplitude of near-field strain oscillation keeps increasing and the process zone size keeps decreasing (Svetlizky et al., JMPS, 2020). Then, I was wondering why quite different rupture behaviors could be observed along the same PMMA fault among different research groups. Note, I don't mean to question the validity of the self-similar rupture behaviors in the authors' study, but just feel puzzled by the seemingly inconsistent results among different studies. Since Nature Communications allows for more space, I suggest discussing such discrepancy and the behind reason(s), so that readers won't get confused.

(1.d) PMMA versus crustal rocks

One can always argue that PMMA is different from typical crustal rocks, so that one must be careful when applying results of PMMA to natural faults. While I am not that picky about this issue, I do feel puzzled (again) by one particular factor — fault roughness. Previous experimental study of Ohnaka and Shen (JGR, 1999) using rocks shows a stabilizing effect of increasing fault roughness. However, here the authors' results using PMMA reveal that increasing fault roughness tends to increase the values of global stress drop and rupture speed (Figures 1 and 3). As fault

roughness is thought to be a key indicator for assessing fault structural maturity and earthquake hazard (Perrin et al., JGR, 2016), I suggest discussing why rougher fault can promote larger stress drop and faster rupture speed, especially considering that the work of Perrin et al. (2016) implies that immature (rough) faults are less likely to host supershear ruptures, which is opposite to the authors' finding.

(1.e) Fracture energy of PMMA

Lines 197-198: Here the authors are referring to some results for steel and other materials (Homalite-100), which may not be analogous to PMMA. Also, loading configuration and fracture/rupture mode (mode-I versus mode-II) are different between previous studies and the current one. On the other hand, Fineberg's group has shown, based on their friction experiments using PMMA, that once loading configuration and interface condition (dry or lubricated) are fixed, the fracture energy is more or less a constant value through stick-slip cycles. Once again, there seem to be an apparent inconsistency among different research groups. Some clarification is needed.

#2 Non-uniqueness for interpreting the results

I agree that self-similar solution is one feasibility that can explain the apparent propagation speed of dynamic ruptures in the sub-Eshelby regime. On the other hand, I feel that the idea of self-similarity is oversold, as implied by the sentences in Lines 47-48, 68-71, 216-218, 255-258, etc., and especially by the use of "unifying" or "unified". I suggest adding some sentences admitting that other alternative mechanisms may also explain natural observations.

#3 The relation to strike-slip faults

In this study, the authors mainly focus on applying their laboratory findings to explaining natural earthquakes along strike-slip faults, but it is not immediately clear why their results are most relevant to strike-slip faults, not dip-slip faults. In fact, some researchers (e.g., Lambert et al., Nature, 2021) have suggested that crack-like ruptures, expected for self-similar ruptures, are more relevant to subduction zone earthquakes, while pulse-like ruptures are more relevant to crustal strike-slip earthquakes. Moreover, as also noticed by the authors, other complexities such as free surface and seismogenic zone width can influence the rupture transition to supershear and the source scaling relation along a strike-slip fault. Considering all of above, I think some clarification is needed to put the laboratory work and natural events in comparably the same framework, so that the authors' argument favoring strike-slip faults can be more meaningful.

#4 The relation between rupture speed and stress drop

Lines 60-62: Here the authors mention that there exists a negative correlation between rupture speed and stress drop. However, later the authors cite another group of natural observations to support a positive correlation between rupture speed and stress drop. Of course, one could argue that there are strong tradeoffs and uncertainties in estimating source properties for natural earthquakes. But for the purpose of this study, if the authors cannot explain the negative correlation reported by refs. 30 and 31, I would recommend not mentioning them in the introduction. Perhaps the authors can mention refs. 30 and 31 in the discussion ("Implications for natural earthquakes"), and briefly discuss the non-uniqueness issue (see comment #2) and the unresolved negative correlation between rupture speed and stress drop.

#5 The definition of static stress drop (inset in Fig. 1f, Fig. 3b)

Previously, some researchers have used the globally measured stress drop to represent static stress drop. However, such defined stress drop can be misleading, as it includes the contributions from both rock (or PMMA) samples and the external machine through the so-called machine stiffness during unloading (Xu et al., Tectophy, 2018; Morad et al., EPSL, 2022). Moreover, each stick-slip event actually can involve repeated ruptures over a total time duration much longer than the propagation time of a single rupture (Fig. 2, Supplementary Fig. 5). Therefore, I suggest adding some sentences to clarify the exact meaning of globally measured stress drop.

#6 Dynamic or static stress drop

During the theoretical analysis, the authors mainly use the dynamic stress drop, estimated during the passage of the rupture front, to "predict" the rupture speed. On the other hand, when referring to the finite fault inversion database, the estimated stress drop most likely represents the static stress drop. Since these two types of stress drop can deviate from one another, I was wondering if such deviation can account for some of the discrepancies seen between laboratory observations and natural observations (refs. 30 and 31). See also my earlier comment #4.

Below is a list of minor comments:

Line 17: It would be helpful to define what Eshelby speed exactly means

Lines 47-48: While unified theory is always attractive, here I don't understand why a unified theory must exist to explain rupture speed stability across all regimes.

Lines 50-54: Several experimental studies (Fukuyama et al., J. Seismol, 2016; Xu et al., Tectophy, 2018; Svetlizky et al., JMPS, 2020) have already shown that rupture can attain a supershear speed immediately following nucleation and can propagate in the conventionally-defined unstable regime, at least for a short period of time. These earlier studies should be cited.

Line 55: Change "outset" to "onset"

Line 90: static shear stress drop cannot be read at the onset of instability

Lines 91-92: I don't recommend calling the globally defined stress ratio "the static friction coefficient", as this may be confusing; rather, please call it "the ratio of global peak shear stress to global normal stress" or something like that. Please also correct the use of terminology in other places, by replacing "f_s" with τ_p/σ_p (e.g., Lines 103, 146-147, 153; Figs. 1e, all panels in Figure 2).

Line 179: Change "Green" to "Yellow"

Line 213: It is not clear how the threshold value "1.08Cs" was obtained. Was it derived from the sophisticated curve fitting procedure (Supplementary Figure 4)? Also, since there are essentially no data in the interval of $C_s < V_r < 1.08 C_s$ (Fig. 4a) or equivalently the interval of $C_s/C_p < \beta < 0.6$ (Supplementary Fig. 4), I suggest downplaying the exact value of "1.08Cs" as the upper boundary for the unstable rupture regime (Lines 213-214, orange belt region in Fig. 4).

Line 279: Change " β " to " α "

Lines 298-299: Could it be possible that some preslip may have been missed, simply because the pre-trigger time window set in the buffer is not long enough?

Section "Self-similar model for subRayleigh and supershear ruptures"

Line 321: Please re-write " G_{ii} " as " G^c_{II} "

Line 323: Please define parameter "k" and the associated condition (plane stress or plane strain)

Line 325: "shear modulus of rigidity" sounds a bit repetitive

Line 356: Change "6.6.92" to "6.9.92"

Whenever possible, please add the corresponding equation number in ref. 44 for all the listed formulae

Line 385: Please define the parameter D (shape function for the cohesive zone model) and describe the assumed distribution within the cohesive zone

Line 418 (Eq. 15b): In the denominator, rupture area "A" is missing

Line 421: A subject is needed before "leads to"

References

- Fukuyama, E., Xu, S., Yamashita, F., and Mizoguchi, K. (2016). Cohesive zone length of metagabbro at supershear rupture velocity. *Journal of Seismology*. 20 (4), 1207–1215.
- Gori, M., Rubino, V., Rosakis, A.J., and Lapusta, N. (2018). Pressure shock fronts formed by ultra-fast shear cracks in viscoelastic materials. *Nature Communications*, 9, 4754.
- Kammer, D.S., Svetlizky, I., Cohen, G., and Fineberg, J. (2018). The equation of motion for supershear frictional rupture fronts. *Science Advances*. 4, eaat5622.
- Lambert, V., Lapusta, N., and Perry, S. (2021). Propagation of large earthquakes as self-healing pulses or mild cracks. *Nature*. 591, 252–258.
- Morad, D., Sagy, A., Tal, Y., and Hatzor, Y.H. (2022). Fault roughness controls sliding instability. *Earth and Planetary Science Letters*. 579, 117365.
- Ohnaka, M., and Shen, L. (1999). Scaling of the shear rupture process from nucleation to dynamic propagation: Implications of geometric irregularity of the rupturing surfaces. *Journal of Geophysical Research*. 104 (B1), 817–844.
- Perrin, C., Manighetti, I., Ampuero, J.P., Cappa, F., and Gaudemer, Y. (2016). Location of largest earthquake slip and fast rupture controlled by along-strike change in fault structural maturity due to fault growth. *Journal of Geophysical Research: Solid Earth*. 121, 3666–3685.
- Shlomai, H., Kammer, D.S., Adda-Bedia, M., and Fineberg, J. (2020). The onset of the frictional motion of dissimilar materials. *The Proceedings of the National Academy of Sciences*, 117 (24) 13379–13385.
- Svetlizky, I., Kammer, D.S., Bayart, E., Cohen, G., and Fineberg, J. (2017). Brittle Fracture Theory Predicts the Equation of Motion of Frictional Rupture Fronts. *Physical Review Letters*. 118, 125501.
- Svetlizky, I., Albertini, G., Cohen, G., Kammer, D., and Fineberg, J. (2020). Dynamic fields at the tip of sub-Rayleigh and supershear frictional rupture fronts. *Journal of the Mechanics and Physics of Solids*. 137, 103826.
- Xu, S., Fukuyama, E., Yamashita, F., Mizoguchi, K., Takizawa, S., and Kawakata, H. (2018). Strain rate effect on fault slip and rupture evolution: Insight from meter-scale rock friction experiments. *Tectonophysics*. 733, 209-231.

Reviewer #2 (Remarks to the Author):

This paper describes some novel and very interesting laboratory experiments on shear ruptures in pre-cut laboratory samples of PMMA. In all cases, apparently, the rupture velocity adopted at the onset of elastodynamics is maintained nearly constant until (?) the rupture fronts reach the edges of the sample. The adopted velocities span the full range between the shear and P-wave speeds, which is not compatible with models of expanding ruptures in which the fracture energy is constant. But these observations do appear to be compatible with observations of natural ruptures that immediately (or as soon as can be observed) attain a super-shear speed.

A simple analysis in this paper shows that constant velocities are consistent with a fracture energy that grows linearly with rupture length, as has been inferred for natural earthquakes, and as has also been predicted by some numerical models of rupture growth when the surrounding medium deforms plastically at a fixed threshold stress level. The authors propose that this rupture velocity is controlled by the ratio $\Delta\tau_d^2/\sigma_N$, where $\Delta\tau_d$ is the “dynamic stress drop” and σ_N is the normal stress, a scaling that causes most of the data to collapse quite well in Fig. 4a (at least for the 2 smoother fault surfaces; experiments are conducted with 3 different fault roughnesses at 3 different “confining” stresses). A popular alternative model, that ruptures start as sub-shear and transition to super-shear when the stress concentration at the shear-wave front spawns a secondary rupture that quickly links with the main rupture (the “Burridge-Andrews mechanism” of this paper), does not explain these data, nor observations of the 2018 M7.5 Palu earthquake (“among several others”, according to the abstract, although these are not named in the main text), which apparently was super-shear from the moment that a velocity could be reliably determined.

The observed rupture velocities in the lab experiments are fit quite well by a theoretically-motivated relation with some free fitting parameters (Fig. 4a). It is not feasible to determine $\Delta\tau_d^2/\sigma_N$ for natural earthquakes, so observed rupture speeds for natural earthquakes are plotted vs the stress drop $\Delta\tau$ as a proxy for $\Delta\tau_d^2/\sigma_N$, as was done for the lab ruptures in Fig. 3b. There is generally a positive trend, qualitatively supporting the arguments presented here.

This is an important paper describing novel experiments that will be of interest to a wide audience. A relatively simple and plausible model describes the data quite well, and the authors show that, at least qualitatively, the concept works for natural earthquakes as well. I think the paper can be published without significant changes. Some of the following suggestions could make this a stronger paper; I think only (1) is really necessary. Although after having written the following, I think that (2) should be addressed at least at a minimal level (providing some guidance for how to interpret Fig. 1e).

(1) Perhaps the most important finding of this paper is that the ruptures rapidly attain a nearly constant velocity, including at a range of velocities between the P- and S-wave speeds. Qualitatively, this seems consistent with the examples shown in Figure 2 and S5. But given its importance, I was surprised to find nearly no discussion of how these velocities were determined, and, perhaps more importantly, no quantitative evaluation of the constancy of those velocities. Is velocity determined from the timing of the stress peaks at the 7 strain gauges in Fig. S5? From differencing the slip measurements in Fig. 2? At several points along the rupture front, or only an average from start to finish? In any case, it seems that it should be possible to estimate the velocity at several points along the rupture path for each event; how constant are those velocities? In Figs. 3 and 4a, each rupture plots at a single velocity with no error bar. I think that a new supplementary figure that shows velocity as a function of propagation distance for an assortment

of ruptures would let the reader judge how constant the rupture velocities are, and what plausible error bars are.

(2) Figures 1b-e show some interesting trends. In all cases except the rough fault with $\sigma_x = 1$ and 2 MPa, the max and min values of $\sigma_y - \sigma_x$ seem to increase monotonically with time or event number. In Fig. 1e, the first ruptures on the “extremely smooth” faults at the different σ_N seem to share a common coefficient of friction of ~ 0.4 . It looks like the same might be said of the smooth fault, but it’s hard to tell. What is most surprising to me is that each color in Fig. 1e (each value of σ_x) follows a very nearly linear trend, with an intercept along the σ_p axis (zero shear strength at a finite normal stress) that seems (coincidentally?) to be very close to σ_x . What does this say about the evolution of the fault surface with successive ruptures? Or could it suggest that, with successive ruptures, the stress along the fault departs from the simple prediction based on the assumed uniform applied load? My understanding is that the shear and normal stress here are determined from the applied loads; can this be checked using the near-fault strain gauges?

In any case, I stared at this figure for a while, wondering how to interpret the x -intercept for each σ_x , before I realized that for the first rupture of a particular fault at each normal stress, the coefficient of friction was well defined. To avoid giving other readers this headache, it would be good to point this out. Although, now that I write this, was there a single manufacturing of each roughness? So that, for example, the same extremely smooth fault was successively subjected to $\sigma_x = 1$ MPa, 2 MPa, and 3 MPa? Then to explain the constant coefficient of friction for this fault for the first events at each normal stress, one would have to imagine that the surface was “reset” somehow (by separating and realigning the plates?). Or might this be evidence of a change in the way the applied loads are transferred to the fault surface? But then this would call into question the relation between rupture speed and $\Delta\tau_d^2/\sigma_N$, because presumably the different values of these parameters come from the stress drops in Fig. 1 that grow with successive rupture events.

(3) The model proposed here assumes that inelastic energy dissipation increases linearly with rupture length. For natural faults, grain-boundary and larger cracks off the fault plane can act as seeds for this deformation. Is it possible to look for signs of permanent off-fault deformation in the PMMA using photoelasticity? Maybe even before and after the first event, to check for damage from the original cutting process, as well as the potential growth of the zone with propagation distance? Or, given the estimated size of the cohesive zone ($d/a \sim 0.04$) and the slip at a distance d behind the rupture front, estimate the near-tip strains and compare this to a plastic failure strength for PMMA? The latter would be much simpler, but still a nice addition.

Minor points:

I don’t recall that it was ever mentioned if the ruptures here were crack-like or pulse-like. From the colors in Fig. 2 it appears that they might be pulse-like, but I suspect that is because the slip contours there saturate at $60 \mu\text{m}$. The stress histories in Fig. S5 don’t obviously show an increase that would be associated with a pulse, but I’m not used to thinking about super-shear slip pulse stress histories. I think the equations in the Methods assume crack-like behavior. Please state whether these ruptures are pulse-like or crack-like, or a mixture; that information must exist.

line 37: “Understanding the controls on rupture speed is thus crucial for the earthquake source mechanisms and thus assessment of seismic hazards.” The “crucial for the earthquake source mechanisms” seems awkward here, and I’m not really sure what it means. How about “crucial for

understanding earthquakes and thus ...”?

line 53: “the Burridge-Andrews mechanism”. For a paper in a general journal this assumes too much knowledge on the part of the reader. Something like “the Burridge-Andrews mechanism, where the jump to super-shear speeds is achieved through the growth of a separate rupture at the shear wave front ahead of the main rupture”, or a separate sentence if that makes the sentence too long. You can probably come up with something better.

line 55: “at the onset”. Presumably there is a length scale below which the observations cannot reach. Something like “from the first moments that a velocity can be discerned” seems safer.

line 61: “observations from natural earthquakes indicate that rupture speed may actually decrease with an increase in the stress drop.” If these observations pertain to super-shear as well as sub-Rayleigh ruptures, this should be stated explicitly.

line 149: When I see “Stress ratio” here, I think of the Burridge-Andrews S , the ratio of the stress differences (peak shear stress at failure minus the ambient shear stress) to (ambient shear stress minus sliding friction), $(\tau_p - \tau_0)/(\tau_0 - \tau_f)$. But here, on line 153, “stress ratio” is identified with f_s , the ratio of the ambient shear stress to normal stress at the onset of slip, or just the static friction. Why not just say “Static friction and shear stress drop” on line 149?

line 155: “However, we observe that the specific relationship between τ_p/σ_p and V_r also varies with σ_x .” σ_x might be the experimentally-controlled variable here, with 3 discrete values, but it really might be σ_p that matters. Have you tried color-coding this figure by the value of σ_p ?

line 168: τ_D is defined as the “dynamic stress drop”. Is that the peak stress at the rupture front minus the quasi-steady sliding friction? Or the ambient shear stress minus the sliding friction? The inset to Fig. 4a suggests that it is the ambient stress minus the sliding friction, in that the small increase in stress just prior to the onset of weakening is ignored (I’m interpreting this increase as due to the arrival of the rupture front). But then how is this different from $\Delta\tau_s$?

line 279: “It can be seen that a rougher fault has a higher β value”. What is β here? Do you mean α ?

line 323: k is defined on line 359, but it should be defined here.

line 356: “equation 6.6.92” should read 6.9.92.

Allan Rubin

Reviewer #3 (Remarks to the Author):

In this manuscript, the authors report an experimental study of interfacial ruptures propagating along a frictional interface at all velocities, from sub-Rayleigh to supershear. The main experimental results are the observation of the forbidden zone for rupture velocity, between the Rayleigh wave velocity and 1.08 of the shear wave velocity, ruptures propagating at constant velocity, and the instantaneous nucleation of shear ruptures without initiation by a sub-Rayleigh rupture. The authors develop an ad-hoc model based on self-similar crack propagation to describe their observations. They are also able to fit seismic data with their model.

The experimental study of the evolution of the velocity of frictional failures is not new and has been solved before for a similar system. However, the observations made in this study are different from those in the literature. In particular, the existence of a band gap has never been explicitly demonstrated on the basis of experimental data. I have some major concerns that I detail below regarding the validity of the observations and measurements. If the authors can convince me of the validity of their observations by providing additional graphs, then the manuscript might be worth publishing in Nature Communications, in my opinion.

Major comments

The dynamics of a frictional rupture has been considered in recent experimental papers (Svetlizky et al [29] then Passelègue et al [26] for sub-Rayleigh ruptures, Kammer et al [45] for supershear ruptures). The authors' claim is that in the present study, constant propagation velocities have been measured. I would like the authors to clarify some of their measurements and observations:

- The main result of the paper and the motivation for deriving a new model is that ruptures propagate at constant velocities. The reader should extract this observation from the space-time plots in Fig. 2. I am not convinced that the velocity of ruptures is indeed constant. I would like the authors to provide the velocity profiles as a function of space for the 9 experiments shown in Fig. 2.
- Assuming that the ruptures do indeed propagate at constant velocity, the stress distributions along the interface need to be clarified. The authors consider that the interface is uniformly loaded based on the fact that nucleation occurs anywhere along the interface. However, the measurements provided by strain gauges must explicitly establish this distribution. Indeed, a non-homogeneous stress distribution can lead to non-uniform rupture dynamics. One possible scenario is that the load decreases along the interface, leading to non-accelerating cracks. But this needs to be checked on the data. I would like the authors to provide the distribution of stresses, both normal and shear, as a function of space - along the interface - prior to rupture propagation.
- From an experimental point of view, the system is made of 20mm-thick blocks. How do the authors ensure that the fracture fronts are straight? If this is not the case, the velocity established from measurements on one face of the blocks will correspond to an effective velocity different from the local and instantaneous velocity, e.g. Rubino et al (Nature 2022, DOI: 10.1038/s41586-022-04749-3). It would lead to a misestimating of the 1.08 C_s limit of the forbidden zone.
- Based on these observations, the authors establish that the LEFM equation of motion is not valid in their experiment. In this paragraph (from l.166), the paper by Passelègue et al [26] is mentioned as establishing the criterion from the LEFM, while it was originally derived and experimentally validated for a 2D system in Svetlizky et al [29]. The explicit citation should be corrected.
- A model is then derived from the self-similar crack model to obtain a new equation of motion of a crack. The derivation appears valid, however, strong assumptions must be made:
 - (i) Increase in fracture energy with fracture length: the authors validate this effect on the basis of seismological data. However, the energy dissipated during a seismic event (G_{EQ}) and the fracture energy used in fracture mechanics (G_{FM}) are not the same quantities. Ke et al (Nat. Comm. 2022, DOI: 10.1038/s41467-022-28647-4) recently showed that by holding G_{FM} constant, they find an increase in G_{EQ} with the rupture length.
 - (ii) Increase in fracture energy with rupture speed: the material tested here is PMMA, as in [29] where no strong dependence of fracture energy on velocity was detected. The references cited on line 331 do not seem to refer to PMMA and to my knowledge there is no such effect in PMMA reported in the literature.

- More generally, in [29] the equation of motion of a crack with constant fracture energy describes the data for all ruptures up to the Rayleigh wave speed, without a need for introducing a dependence both on length and velocity. Can the authors comment on the origin of the differences in behavior in these two very similar systems?
- In addition to these two assumptions, the model involves rewriting the expression for $g_{II}(v)$ for a self-similar crack instead of a steady crack. However, the two expressions are very similar especially for intermediate velocities. I would like the authors to provide (i) a quantitative comparison of the results from the two models – self-similar and steady, and (ii) a comparison of their measurements with the model derived for a steady rupture.
- Eventually, the logic of the study seems to me to be flawed in one respect. The authors derive a different model from LEFM because it cannot predict constant velocity propagation with constant fracture energy. However, later the authors consider a non-constant fracture energy to describe their data (Eqs. 4 & $D_c = \eta a$). With these assumptions, LEFM could describe the data as well as the self-similar crack.
- The paper extends the model to supershear fractures. How does it compare to other more realistic than self-similar crack models, such as the solution of Huang and Gao (JAM 2001, DOI: 10.1115/1.1357871)?
- The authors report qualitative observations on the effect of surface roughness on interface stability and strength. Can they provide an interpretation of these observations?

Minor comments

- lines 67-68: “The rupture speed is dictated by the driving load that depends on the dynamic stress drop and the normal stress”. The stress drop depends on the driving load and not the opposite.
- line 89: The definition of the peak shear stress and of the static shear stress drop should be clarified and defined on the inset of Fig. 4a, as $\Delta\tau_d$. Does it correspond to the initial stress applied to the fault when the rupture nucleates or to the contact shear strength? The latter cannot be measured directly.
- lines 183-184: The writing is confusing. “In this model fracture speed is constant... The fracture energy is thus constant in this model”... “an expanding fracture has to accelerate”. It should be corrected.
- line 205: Note that speeds between C_R and $\sqrt{2}C_s$ have been measured in Svetlizky et al (JMPS 2020, DOI: 10.1016/j.jmps.2019.103826) and Kammer et al [45].
- line 356: the reference to section 6.6.92 should read 6.9.92.

RESPONSE TO REVIEWERS' COMMENTS

Review comments are shown in plain text, responses in blue, and changes in red in the revised **Manuscript and Supplemental Information**.

Comments from reviewer

Reviewer #1:

General comments: By simulating spontaneous dynamic ruptures along a polymethyl methacrylate (PMMA) laboratory fault, Dong et al. provide new insights into the stability of rupture speed in both the sub-Rayleigh and supershear regimes. On the basis of self-similar solutions, they further identify a universal “driving force” that can fit and explain their experimental observations. Finally, the authors link their results to natural observations, showing that their analysis can also explain the rupture speed and source scaling relation for a suite of natural earthquakes along strike-slip faults. I find this manuscript interesting and suitable for publication in Nature Communications. It represents an excellent piece of work that combines experiments, theory, and natural observations. I also think the results will attract a wide range of readers across geoscience, material science, natural hazard, etc.

In the meantime, I do have some concerns, mostly on the use of PMMA and the comparison to several previous studies. I don't think these concerns will affect the overall analysis or conclusion of this work, but just feel that some clarification and discussion are needed to reinforce the reasoning. After these concerns are properly addressed, I would be happy to make my further recommendation.

Response: Thanks for the precise summary of our work. Your recognition of the impact of this work is greatly appreciated. We address these concerns to reinforce our reasoning as you suggested. Your main concern in the use of PMMA is addressed in **Response #1**. As you note, we updated the P wave velocity from the bulk to plane stress. The data trends and main conclusions remain unchanged. In addition, we note one possible explanation for the difference in experimental observations between this current work and studies from Jay Fineberg's group: the stress state along the laboratory fault. Following Scholz et al. (1972), we utilized biaxial loading to achieve a more uniform stress along the fault as compared with other loading methods for laboratory earthquakes.

Major comments:

#1 The use of PMMA

(1.a) Previous studies show that PMMA has a viscoelastic behavior, whose effective elastic moduli (or wave velocities) could increase with the local strain rate (Gori et al., Nature Comm, 2018) (Gori et al., 2018). Since the exact values of wave velocities are important for defining the stability regimes of dynamic ruptures, I was wondering whether the authors have performed an accurate measurement of wave velocities for PMMA in the dynamic range, as done by Shlomai et al. (PNAS, 2020).

Response: We agree that PMMA is viscoelastic and that measurement of wave velocities is important for defining the stability regimes. In this study, we used the ultrasonic method to quantify the wave velocities of the PMMA – similar to Shlomai et al. (2020). We obtained a similar S wave velocity, but our P wave velocity corresponds to the bulk P wave velocity. In the revision, we changed the P wave velocity from bulk (~2.6 km/s) to plane stress (~2.4 km/s) and revised the other parameters in the model accordingly. These modifications did not change the trends in observations and overall conclusions.

(1.b) Plane stress or plane strain

I may have missed it, but I find the information of plane stress or plane strain is never clearly mentioned in the text. This information not only affects the effective P wave velocity for a fault with finite width, but also influences the value of parameter “k” in the “Self-similar model for subRayleigh and supershear ruptures” section in Methods. I therefore suggest adding the assumed condition for plane stress or plane strain explicitly.

Response: Indeed, you are correct. Plane stress prevails in our experiments. Accordingly, we explicitly clarify the stress condition in the revision as:

306 km/s and $C_p=2.40$ km/s for plane stress conditions.

and:

427 for a static mode II rupture with $\Delta\tau_d$ being the dynamic stress drop, k the ratio between C_s and C_p

428 for plane stress conditions, a the crack half length, and μ the shear modulus.

(1.c) Transient versus self-similar rupture behaviors

Jay Fineberg's group has studied extensively the behaviors of laboratory earthquakes along a PMMA fault. Their results generally show a monotonically increasing trend of rupture speed towards some limit value, for both the sub-Rayleigh (Svetlizky et al., PRL, 2017) and the supershear regimes (Kammer et al., ScienceAdv, 2018; Svetlizky et al., JMPS, 2020). Although the asymptotic rupture speed in their studies does approach a constant value in each regime, the overall rupture behavior does not follow the strict definition for steady-state solution or self-similar solution, because the amplitude of near-field strain oscillation keeps increasing and the process zone size keeps decreasing (Svetlizky et al., JMPS, 2020). Then, I was wondering why quite different rupture behaviors could be observed along the same PMMA fault among different research groups. Note, I don't mean to question the validity of the self-similar rupture behaviors in the authors' study, but just feel puzzled by the seemingly inconsistent results among different studies. Since Nature Communications allows for more space, I suggest discussing such discrepancy and the behind reason(s), so that readers won't get confused.

Response: Thanks. Indeed, Jay Fineberg's group has contributed significantly to the understanding of laboratory earthquakes along a PMMA fault. At the same time, we agree with your observation that in their studies "the overall rupture behavior does not follow the strict definition for a steady-state or self-similar solution". It is exactly the objective of this work to conduct experiments that take advantage of either steady-state or self-similar solution by ensuring a uniform stress distribution along the fault as detailed below (also noted above).

In the previous studies from Jay Fineberg's group, the experiments were conducted using a direct shear apparatus. Due to the moment produced in the specimen by the loading configuration, the normal stress and shear stress are likely nonuniform as first pointed out by Scholz et al. (1972) and explained here. Considering the free-body diagram of the upper half of the plate in a direct shear experiment (**Fig. R1**), the frictional stress acts on the fault, and the normal stress is uniformly distributed along the upper boundary. We can then calculate the torque with respect to point A, the center of the laboratory fault. Both the frictional stress along the fault and the normal stress at the boundary do not contribute to the torque relative to this point. However, the external shear force causes a clockwise torque (T_1) with respect to this point. According to the equilibrium condition, the normal stress on the laboratory fault must generate a counterclockwise torque (T_2). Therefore,

the normal stress on the interface must be nonuniform, increasing from the loading edge to the leading edge (**Fig. R1**). The rupture therefore tends to initiate from the loading side where the normal stress and thus the frictional resistance is lower.

The above nonuniformity has been confirmed by experimental observation of the rupture initiation point from Jay Fineberg's group. With reference to their setup (see **Fig. 1** of Ben-David et al. (2010)), the rupture often initiates from the low normal stress point (the loading edge of the top block) and propagates towards the high stress region (Kammer et al., 2018; Svetlizky et al., 2020; Svetlizky et al., 2017). Further because of the gradient of the normal stress and thus likely the stress drop, the rupture tends to accelerate during propagation. Therefore, it is rare to observe rupture with nearly constant propagation velocity in their experiments.

Fig. R1. Schematic diagram shows the nonuniform distribution of normal stress along the laboratory fault in direct shear experiments.

In this work, the laboratory fault model is biaxially loaded following Scholz (Scholz et al., 1972; Scholz, 2002), producing a relatively uniform stress along the fault. We used four loading platens moving in a synchronized manner to maximize the uniformity of stress along the fault. We have added **Supplementary Fig. 4** to the manuscript to demonstrate this point - to show the distribution of shear stress and dynamic stress drop along the fault in three sequences of stick-slip events. Given the large dimension of the specimen, the shear stress before the onset of rupture is relatively uniform, especially in the central segment of the laboratory fault where ruptures initiate and propagate. In particular, the distributions of shear stress and dynamic stress drop are clearly more uniform than those in direct shear experiments (Ben-David et al., 2010). What is more interesting

from the figure is that the distribution of dynamic stress drop is more uniform than the distribution of initial shear stress. We believe that the dynamic stress drop measurement is more reliable because it is transient and thus less affected by the measurement errors due to thermal drift.

We fully respect outstanding contributions on laboratory earthquakes from Jay Fineberg's group. The current study is mainly motivated by their findings. The necessarily non-uniform stress distribution consequent of their direct shear configuration may have contributed to the observation of non-steady rupture speed in their results. We added our interpretation in the revision as:

mechanisms for supershear ruptures are possible³⁹⁻⁴¹. Furthermore, nearly steady subEshelby
supershear speeds are observed (Supplementary Figs. 5 and 6), which confirms the other puzzle
in the 2018 Palu earthquake. Although rupture speeds between C_S and C_E have been reported
149 ^{38,42}, the observed rupture speeds are always transient. These observations of non-steady rupture
speed may be associated with the stress gradient along the fault in direct shear experiments²⁶ as
first pointed out by Scholz^{43,44}.

(1.d) PMMA versus crustal rocks

One can always argue that PMMA is different from typical crustal rocks, so that one must be careful when applying results of PMMA to natural faults. While I am not that picky about this issue, I do feel puzzled (again) by one particular factor — fault roughness. Previous experimental study of Ohnaka and Shen (JGR, 1999) using rocks shows a stabilizing effect of increasing fault roughness. However, here the authors' results using PMMA reveal that increasing fault roughness tends to increase the values of global stress drop and rupture speed (Figures 1 and 3). As fault roughness is thought to be a key indicator for assessing fault structural maturity and earthquake hazard (Perrin et al., JGR, 2016), I suggest discussing why rougher fault can promote larger stress drop and faster rupture speed, especially considering that the work of Perrin et al. (2016) implies that immature (rough) faults are less likely to host supershear ruptures, which is opposite to the authors' finding.

Response: We appreciate this comment that gives us the opportunity to further consider the effect of roughness on fault rupture behaviors. We used the finding from a recent study by Morad et al. (2022) to justify the discrepancy between our observations and those by Ohnaka and Shen (1999)

and Perrin et al. (2016). We added a discussion on the effect of roughness on rupture dynamics in the revision to clarify, as:

**Discussion**

Mature faults with large slip offsets are typically smoother than short-offset immature faults
⁵⁵. Congruently, high rupture speed is correlated with mature fault sections ⁵⁶. This is
qualitatively consistent with previous experimental observations that increasing fault roughness
has a stabilizing effect on fault rupture ⁵⁷ - seemingly contrary to our findings, here. This
discrepancy may arise from the limited range of roughness involved in this study. Recently, the
effect of fault roughness on rupture dynamics has been resolved over a wide range of
roughnesses ⁵⁸. This has defined a critical roughness, below which the stress drop increases with
roughness, and beyond which the stress drop decreases with roughness. This suggests that an
increase in roughness may not necessarily stabilize fault rupture. In this work, the fault is overall
smooth (Supplementary Fig. 1). Thus, an increase in fault roughness may promote fault
instability according to observations by Morad, et al. ⁵⁸. Our results are generally consistent with
those of Xu, et al. ³¹, who demonstrated that rough faults could indeed accommodate supershear
rupture.

(1.e) Fracture energy of PMMA

Lines 197-198: Here the authors are referring to some results for steel and other materials (Homalite-100), which may not be analogous to PMMA. Also, loading configuration and fracture/rupture mode (mode-I versus mode-II) are different between previous studies and the current one. On the other hand, Fineberg's group has shown, based on their friction experiments using PMMA, that once loading configuration and interface condition (dry or lubricated) are fixed, the fracture energy is more or less a constant value through stick-slip cycles. Once again, there seem to be an apparent inconsistency among different research groups. Some clarification is needed.

Response: Yes, thanks for the opportunity to clarify. We agree with you that the material and loading configuration are different between this work and the referred studies. The dependence of fracture energy on rupture velocity should thus be considered as a general assumption for data

analysis. The exact form of this dependence can be obtained from experimental results in an *ad hoc* manner.

Our results are not necessarily contrary to those from the Fineberg group. First, the stress distribution along the fault in the current work is different from that in their studies as explained in the response to your **Comment #1**. Thus, direct comparison of results on fracture energy may not be fully meaningful. Secondly, the proportionality between the **specific fracture energy** and the rupture length is a theoretical consequence of the self-similar rupture model. It is emphasized here that “specific fracture energy” is a generalized concept for energy dissipation that may include fracture energy at the rupture tip, off-fault yielding, and spatially extended dissipation (Broberg, 2002; Templeton, 2009). In the original manuscript, we used “fracture energy” to represent the sum of the ensemble energy dissipation. To avoid confusion, we call the total energy dissipated as “specific fracture energy” following Templeton (2009) in the revision. We added the clarification in the revision as:

Rupture dynamics is governed by energy balance. For mode II shear rupture with constant
specific fracture energy (i.e., total energy dissipation), the rupture will accelerate towards the
limiting speed C_R , and some of them could suddenly jump into supershear regime with the final
speed approaching the other limiting speed C_p ²⁰. However, a constant rupture speed has been
observed in seismological observations¹⁷. This constant rupture speed requires an increase in
dissipated energy with growing rupture length⁵². Several mechanisms have been proposed to
provide this increase, including off-fault yielding⁵², provision of specific friction laws⁶⁸⁻⁷⁰, and
rupture speed-dependent fracture energy⁷¹. Experiments have revealed that for constant rupture
speed, the dissipated energy increases with rupture length, at least for mode I ruptures⁵⁴.

In a self-similar rupture model, an inherent theoretical consequence is that the specific
fracture energy Γ_{II} must increase with the rupture length⁷². Γ_{II} is a generalized concept that may
contain energy consumed at the rupture tip (fracture energy), inelastic dissipation and spatially
extended dissipation⁵². The proportionality between Γ_{II} and rupture length is consistent with
dynamic rupture models with off-fault dissipation^{49,52,72} and non-edge-localized dissipation⁶⁸.
Also, this proportionality is consistent with field observations that “seismological” fracture
energy (breakdown work) scales with the rupture size⁷³.

#2 Non-uniqueness for interpreting the results

I agree that self-similar solution is one feasibility that can explain the apparent propagation speed of dynamic ruptures in the subEshelby regime. On the other hand, I feel that the idea of self-similarity is oversold, as implied by the sentences in Lines 47-48, 68-71, 216-218, 255-258, etc., and especially by the use of “unifying” or “unified”. I suggest adding some sentences admitting that other alternative mechanisms may also explain natural observations.

Response: Thanks. We should clarify that “unifying” or “unified” indicates that the self-similar model works for ruptures accommodating speeds from subRayleigh through supershear regimes. Nevertheless to avoid confusion, we revise these sentences following your suggestions as:

- From:

ruptures involving oblique slip¹⁸. However, these models all require special conditions and are
incapable of interpreting stable subRayleigh speeds as well in a unified way.

to:

However, these models all require special conditions and are incapable of interpreting stable
subRayleigh speeds.

- From:

strain energy available to drive the rupture propagation. A unified self-similar rupture model for
earthquake dynamic rupture is developed. The model prediction of rupture speed stability ranges

to:

strain energy available to drive the rupture propagation. A self-similar rupture model covering
speeds from subRayleigh to supershear is developed. The model prediction of rupture speed

- From:

field observation by invoking geometrically sophisticated fault models^{15,16,18}, we provide an
elegant and unified resolution here.

to:

geometrically sophisticated fault models^{16,18,20}, we provide an elegant and simple resolution
here.

- From:

importance in the interpretation of the spectrum of earthquake rupture speeds, in defining critical
controls of rupture speeds, in revealing the unified nucleation mechanism, and therefore in the
assessment of seismic hazards.

to:

294 spectrum of earthquake rupture speeds, in defining critical controls of rupture speeds, in
295 revealing nucleation mechanisms, and therefore in the assessment of seismic hazards. Because

We also add the following sentence in the revision as

**revealing nucleation mechanisms**, and therefore in the assessment of seismic hazards. **Because**
**natural faults are more heterogeneous than our laboratory faults, other alternative mechanisms**
**may also exist under particular circumstances.**

#3 The relation to strike-slip faults

In this study, the authors mainly focus on applying their laboratory findings to explaining natural earthquakes along strike-slip faults, but it is not immediately clear why their results are most relevant to strike-slip faults, not dip-slip faults. In fact, some researchers (e.g., Lambert et al., Nature, 2021) have suggested that crack-like ruptures, expected for self-similar ruptures, are more relevant to subduction zone earthquakes, while pulse-like ruptures are more relevant to crustal strike-slip earthquakes. Moreover, as also noticed by the authors, other complexities such as free surface and seismogenic zone width can influence the rupture transition to supershear and the source scaling relation along a strike-slip fault. Considering all of above, I think some clarification is needed to put the laboratory work and natural events in comparably the same framework, so that the authors' argument favoring strike-slip faults can be more meaningful.

Response: We agree with you that it would be nice to explore more applications of the current study. However, constrained by the geometry of the laboratory fault (thickness of 20 mm and length of ~700 mm), the rupture is approximately mode II rupture. We clarify this fact in the revision as:

Finally, we compare our model with observations of natural earthquakes. **Generally, strike-**
**slip earthquakes on elongated faults may be simplified as mode-II shear ruptures – replicated by,**
**and most relevant to, our laboratory geometry.** Since determining the absolute normal stress for

#4 The relation between rupture speed and stress drop

Lines 60-62: Here the authors mention that there exists a negative correlation between rupture speed and stress drop. However, later the authors cite another group of natural observations to support a positive correlation between rupture speed and stress drop. Of course, one could argue

that there are strong tradeoffs and uncertainties in estimating source properties for natural earthquakes. But for the purpose of this study, if the authors cannot explain the negative correlation reported by refs. 30 and 31, I would recommend not mentioning them in the introduction. Perhaps the authors can mention refs. 30 and 31 in the discussion (“Implications for natural earthquakes”), and briefly discuss the non-uniqueness issue (see comment #2) and the unresolved negative correlation between rupture speed and stress drop.

Response: Agreed. We revise the manuscript following your suggestion by moving the related statements into the section of Discussion of the revision as:

results. We also note that some observations from natural earthquakes indicate that rupture speed
may decrease with an increase in the stress drop, especially for earthquakes with subRayleigh
rupture speeds^{60,61}. However, the validity of this finding may suffer from tradeoffs and
uncertainties in estimating source properties for natural earthquakes.

#5 The definition of static stress drop (inset in Fig. 1f, Fig. 3b)

Previously, some researchers have used the globally measured stress drop to represent static stress drop. However, such defined stress drop can be misleading, as it includes the contributions from both rock (or PMMA) samples and the external machine through the so-called machine stiffness during unloading (Xu et al., Tectophy, 2018; Morad et al., EPSL, 2022). Moreover, each stick-slip event actually can involve repeated ruptures over a total time duration much longer than the propagation time of a single rupture (Fig. 2, Supplementary Fig. 5). Therefore, I suggest adding some sentences to clarify the exact meaning of globally measured stress drop.

#6 Dynamic or static stress drop

During the theoretical analysis, the authors mainly use the dynamic stress drop, estimated during the passage of the rupture front, to “predict” the rupture speed. On the other hand, when referring to the finite fault inversion database, the estimated stress drop most likely represents the static stress drop. Since these two types of stress drop can deviate from one another, I was wondering if such deviation can account for some of the discrepancies seen between laboratory observations and natural observations (refs. 30 and 31). See also my earlier comment #4.

Response: Thanks for the thoughtful comments. We agree that a conventional definition of static stress drop may be misleading. In this work, we define the globally measured stress drop as the static stress drop. We also agree that each stick-slip event actually can involve repeated ruptures, as evident in three typical events in the added **Supplementary Fig. 5**. We confirm that in the finite fault inversion database, the estimated stress drop represents the static stress drop. We discuss the difference between dynamic stress drop and static stress drop in the newly added section **Static and dynamic stress drop** in the revision:

**Static and dynamic stress drop**

In this work, static stress drop $\Delta\tau_s$ is defined as the global drop in the macroscopic shear
stress resolved on the fault (inset in Fig. 1f) - similar to previous studies⁶⁷. However, static stress
drop may not accurately quantify the change in shear stress associated with the rupture. This is
because static stress drop includes the contributions from both the specimen and the load frame
“machine stiffness” during unloading^{37,58}. Furthermore, static stress drop is an integration of
stress change resulting from multiple ruptures during a stick-slip event. The free boundary at the
end of the fault is subject to surface breakout when a rupture arrives³¹, potentially allowing
backward-propagating re-ruptures, as shown in Supplementary Fig. 5. Thus, the globally
measured static stress drop $\Delta\tau_s$ cannot represent the stress drop accompanying the first rupture
during a stick-slip event. To isolate the stress change of the first rupture from the global value,
we measure the dynamic stress drop $\Delta\tau_d$ of the first rupture from the evolution of the shear
stress, as shown in the inset in Fig. 4a. This treatment is in accordance with previous studies^{30,37}.

For natural earthquakes, because the faults are embedded in a nearly infinite medium (the
crust), boundary-induced repeated ruptures are likely absent. Thus, the dynamic stress drop $\Delta\tau_d$
of laboratory earthquakes is equivalent to the “seismological” stress drop $\Delta\tau$ in the investigation
of natural earthquakes, especially for ruptures on simple faults.

Minor comments:

1. Line 17: It would be helpful to define what Eshelby speed exactly means

Response: Thanks. We revise the manuscript according to this kind suggestion as follows:

16 accelerate to the Rayleigh wave speed with some of them further jumping to stable
17 supershear speeds above the Eshelby speed ($\sim\sqrt{2}$ of the shear wave speed). However, the

2. Lines 47-48: While unified theory is always attractive, here I don't understand why a unified theory must exist to explain rupture speed stability across all regimes.

Response: Agreed that it is not necessary for a model to explain rupture speed stability across all regimes in a unified way. We revise this sentence in the revision as:

49 However, these models all require special conditions and are incapable of interpreting stable
50 subRayleigh speeds.

3. Lines 50-54: Several experimental studies (Fukuyama et al., J. Seismol, 2016; Xu et al., Tectophy, 2018; Svetlizky et al., JMPS, 2020) have already shown that rupture can attain a supershear speed immediately following nucleation and can propagate in the conventionally-defined unstable regime, at least for a short period of time. These earlier studies should be cited.

Response: We appreciate this comment. We cite these references in the revision:

the literature featuring numerical or experimental abrupt triggering. Such immediate transition to
supershear has also been observed from some experiments³⁶⁻³⁸, but with limited resolution. In

and:

in the 2018 Palu earthquake. Although rupture speeds between C_S and C_E have been reported
149 ^{38,42}, the observed rupture speeds are always transient. These observations of non-steady rupture

4. Line 55: Change "outset" to "onset"

Response: Thanks. We revise the sentence in the revision as:

59 However, the 2018 Mw 7.5 Palu earthquake attains supershear speed at the onset (or from the
60 first moments that a velocity can be discerned)^{14,25}, challenging such conventional

5. Line 90: static shear stress drop cannot be read at the onset of instability

Response: Thanks. We revise the sentence in the revision as:

For each event, we calculate the global peak shear stress (τ_p) and normal stress (σ_p) at the
onset of instability by resolving σ_x and σ_y on the fault plane. Experimental results show that τ_p

6. Lines 91-92: I don't recommend calling the globally defined stress ratio "the static friction coefficient", as this may be confusing; rather, please call it "the ratio of global peak shear stress to global normal stress" or something like that. Please also correct the use of terminology in other places, by replacing "f_s" with τ_p/σ_p " (e.g., Lines 103, 146-147, 153; Figs. 1e, all panels in Figure 2).

Response: Thanks. We refer to the defined stress ratio as "the ratio of global peak shear stress to normal stress (τ_p/σ_p)" and correct the terminology in Fig. 1e and Fig. 2 and in the revision as:

90 For each event, we calculate the global peak shear stress (τ_p) and normal stress (σ_p) at the
91 onset of instability by resolving σ_x and σ_y on the fault plane. Experimental results show that τ_p

global peak shear stress τ_p as a function of peak normal stress σ_p , at the onset of instability for each event. Black
dashed lines correspond to different ratios of τ_p/σ_p . f, Static shear stress drop $\Delta\tau_s$ as a function of peak normal

**Fig. 2. Nucleation and rupture process of spontaneous events.** Each subgraph shows the space-time distribution
of fault parallel slip. Colors indicate the distributed measurements of fault slip obtained from optical images. The
magenta lines correspond to the shear stress measured using the shear strain gauges. The ratio of global peak shear
stress to normal stress (τ_p/σ_p) and the measured rupture speed V_r normalized by shear wave speed C_s (see Methods)
of each event are indicated on each subgraph.

7. Line 179: Change "Green" to "Yellow"

Response: Thanks. We revise the sentence in the revision as:

smooth, and rough faults are represented by circles, squares, and triangles, respectively. Yellow, orange and red
indicate events at horizontal stresses $\sigma_x = 1$ MPa, 2 MPa, and 3 MPa, respectively.

8. Line 213: It is not clear how the threshold value “1.08Cs” was obtained. Was it derived from the sophisticated curve fitting procedure (Supplementary Fig. 4)? Also, since there are essentially no data in the interval of $C_s < V_r < 1.08 C_s$ (Fig. 4a) or equivalently the interval of $C_s/C_p < \beta < 0.6$ (Supplementary Fig. 4), I suggest downplaying the exact value of “1.08Cs” as the upper boundary for the unstable rupture regime (Lines 213-214, orange belt region in Fig. 4).

Response: Thanks. Yes, the threshold value was derived from the fitting procedure. We agree that it is better to downplay the precision of the exact value of “1.08Cs”. Thus, in the revision we defined a velocity of $C_U(\sim 1.08C_S)$ as:

**Stability of dynamic ruptures.** These same data (Fig. 4a) show stable rupture speeds from
$\sim 0.2C_S$ to near C_R and from $C_U(\sim 1.08C_S)$ up to C_P , consistent with field observations²⁰.

9. Line 279: Change “\beta” to “\alpha”

Response: Thanks. We correct the typo in the revision as:

smooth, and rough fault, respectively. It can be seen that a rougher fault has a higher α value,
consistent with existing studies⁵⁶.

10. Lines 298-299: Could it be possible that some preslip may have been missed, simply because the pre-trigger time window set in the buffer is not long enough?

Response: We appreciate this comment. We agree that some preslip may have been missed because of the short pre-trigger time window. Preslip may develop several seconds before the instability (Ohnaka and Shen, 1999). In this work, the observed preslip is at the final stage, 2 ms before the instability.

Section “Self-similar model for subRayleigh and supershear ruptures”

11. Line 321: Please re-write “ $G_{(ii)}$ ” as “ $G^c_{(II)}$ ”

Response: We appreciate the comment. $G_{(ii)}$ refers to the energy release rate while $G^c_{(II)}$ refers to the specific fracture energy. In the revision, $G^c_{(II)}$ is renamed $G_{(II)}$ to avoid confusion.

12. Line 323: Please define parameter “ k ” and the associated condition (plane stress or plane strain)

Response: Indeed. We correct the sentence in the revision as:

for a static mode II rupture with $\Delta\tau_d$ being the dynamic stress drop, k the ratio between C_S and C_P
for plane stress conditions, a the crack half length, and μ the shear modulus.

13. Line 325: “shear modulus of rigidity” sounds a bit repetitive

Response: We appreciate this comment. We correct the sentence in the revision as:

428 for plane stress conditions, a the crack half length, and μ the shear modulus.

14. Line 356: Change “6.6.92” to “6.9.92”

Response: We appreciate the comment and change the text accordingly.

For a subRayleigh rupture, the analytical expression for $g_{II}(\beta)$ is (equation 6.9.92 in reference
⁵¹)

15. Whenever possible, please add the corresponding equation number in ref. 44 for all the listed formulae

Response: We appreciate the comment and make the change accordingly.

cohesive zone at the crack tip (equation 6.9.124 in reference ⁵¹ and equation 68 in reference ⁷⁶) is

16. Line 385: Please define the parameter D (shape function for the cohesive zone model) and describe the assumed distribution within the cohesive zone.

Response: Thanks. We add the function in the revision as:

$$488 \quad D(-\xi) = 1 - \xi$$

The type of cohesive zone is clarified in the revision as:

475 when $V_r = C_E$, the Eshelby velocity. By introducing a **Barenblatt-type** cohesive zone, the energy

17. Line 418 (Eq. 15b): In the denominator, rupture area “A” is missing

Response: We appreciate this comment and correct the equation in the revision as:

$$521 \quad \Delta\tau = C \left(\frac{L_r}{W_r} \right) \frac{M_0}{AW_r} \quad L_r > W_r \quad (15b)$$

18. Line 421: A subject is needed before “leads to”

Response: We appreciate this comment and make changes in the revision as:

Noting that the rupture dimension in SRCMOD is always larger than the rupture area on the
real fault. **This oversized rupture dimension** leads to an underestimate of stress drop when using

References:

- Ben-David, O., Cohen, G., & Fineberg, J. (2010), The dynamics of the onset of frictional slip. *Science*, 330(6001), 211-214.
- Broberg, K. B. (2002), Constant velocity crack propagation—dependence on remote land. *International Journal Of Solids And Structures*, 39, 6403-6410.
- Gori, M., Rubino, V., Rosakis, A. J., & Lapusta, N. (2018), Pressure shock fronts formed by ultra-fast shear cracks in viscoelastic materials. *Nature Communications*, 9(1), 4754.
- Kammer, D. S., Svetlizky, I., Cohen, G., & Fineberg, J. (2018), The equation of motion for supershear frictional rupture fronts. *Science Advances*, 4(7), eaat5622.
- Morad, D., Sagy, A., Tal, Y., & Hatzor, Y. H. (2022), Fault roughness controls sliding instability.

- Earth And Planetary Science Letters*, 579.
- Ohnaka, M., & Shen, L.-F. (1999), Scaling of the shear rupture process from nucleation to dynamic propagation: Implications of geometric irregularity of the rupturing surfaces. *Journal of Geophysical Research*, 104(B1), 817-844.
- Perrin, C., Manighetti, I., Ampuero, J.-P., Cappa, F., & Gaudemer, Y. (2016), Location of largest earthquake slip and fast rupture controlled by along-strike change in fault structural maturity due to fault growth. *Journal of Geophysical Research: Solid Earth*, 121, 3666-3685.
- Scholz, C., Molnar, P., & Johnson, T. (1972), Detailed studies of frictional sliding of granite and implications for the earthquake mechanism. *Journal of Geophysical Research*, 77(32), 6392-6406.
- Scholz, C. H., 2002. *The Mechanics of Earthquakes and Faulting*, 2nd ed. Cambridge University Press, New York.
- Shlomai, H., Kammer, D. S., Adda-Bedia, M., & Fineberg, J. (2020), The onset of the frictional motion of dissimilar materials. *Proceedings of the National Academy of Sciences of the United States of America*, 117(24), 13379-13385.
- Svetlizky, I., Albertini, G., Cohen, G., Kammer, D. S., & Fineberg, J. (2020), Dynamic fields at the tip of sub-Rayleigh and supershear frictional rupture fronts. *Journal of the Mechanics and Physics of Solids*, 137, 103826.
- Svetlizky, I., Kammer, D. S., Bayart, E., Cohen, G., & Fineberg, J. (2017), Brittle Fracture Theory Predicts the Equation of Motion of Frictional Rupture Fronts. *Physical Review Letters*, 118(12), 125501.
- Templeton, E., 2009. *Effects of Inelastic Off-Fault Deformation on the Dynamics of Earthquake Rupture and Branch Fault Activation*. Harvard University.

Reviewer #2 Allan Rubin (Remarks to the Author):

This paper describes some novel and very interesting laboratory experiments on shear ruptures in pre-cut laboratory samples of PMMA. In all cases, apparently, the rupture velocity adopted at the onset of elastodynamics is maintained nearly constant until (?) the rupture fronts reach the edges of the sample. The adopted velocities span the full range between the shear and P-wave speeds, which is not compatible with models of expanding ruptures in which the fracture energy is constant. But these observations do appear to be compatible with observations of natural ruptures that immediately (or as soon as can be observed) attain a super-shear speed.

A simple analysis in this paper shows that constant velocities are consistent with a fracture energy that grows linearly with rupture length, as has been inferred for natural earthquakes, and as has also been predicted by some numerical models of rupture growth when the surrounding medium deforms plastically at a fixed threshold stress level. The authors propose that this rupture velocity is controlled by the ratio $(\Delta\tau_d)^2/\sigma_N$, where $\Delta\tau_d$ is the “dynamic stress drop” and σ_N is the normal stress, a scaling that causes most of the data to collapse quite well in Fig. 4a (at least for the 2 smoother fault surfaces; experiments are conducted with 3 different fault roughnesses at 3 different “confining” stresses). A popular alternative model, that ruptures start as sub-shear and transition to super-shear when the stress concentration at the shear-wave front spawns a secondary rupture that quickly links with the main rupture (the “Burridge-Andrews mechanism” of this paper), does not explain these data, nor observations of the 2018 M7.5 Palu earthquake (“among several others”, according to the abstract, although these are not named in the main text), which apparently was super-shear from the moment that a velocity could be reliably determined

The observed rupture velocities in the lab experiments are fit quite well by a theoretically-motivated relation with some free fitting parameters (Fig. 4a). It is not feasible to determine $(\Delta\tau_d)^2/\sigma_N$ for natural earthquakes, so observed rupture speeds for natural earthquakes are plotted vs the stress drop $\Delta\tau$ as a proxy for $(\Delta\tau_d)^2/\sigma_N$, as was done for the lab ruptures in Fig. 3b. There is generally a positive trend, qualitatively supporting the arguments presented here.

This is an important paper describing novel experiments that will be of interest to a wide audience. A relatively simple and plausible model describes the data quite well, and the authors

show that, at least qualitatively, the concept works for natural earthquakes as well. I think the paper can be published without significant changes. Some of the following suggestions could make this a stronger paper; I think only (1) is really necessary. Although after having written the following, I think that (2) should be addressed at least at a minimal level (providing some guidance for how to interpret Fig. 1e).

Response: We appreciate your positive comments. Addressing your first comment, we add a section in **Methods** to introduce how the rupture velocity is determined. We also add two figures (**Supplementary Figs. 5 and 6**) to demonstrate and emphasize the nearly constant rupture speed observed along the fault. The error in rupture velocity is also estimated. Addressing your second comment, we explain the results in detail – hopefully this clarifies issues in the analysis and interpretation of these results.

Major comments:

#1 Perhaps the most important finding of this paper is that the ruptures rapidly attain a nearly constant velocity, including at a range of velocities between the P- and S-wave speeds. Qualitatively, this seems consistent with the examples shown in Figure 2 and S5. But given its importance, I was surprised to find nearly no discussion of how these velocities were determined, and, perhaps more importantly, no quantitative evaluation of the constancy of those velocities. Is velocity determined from the timing of the stress peaks at the 7 strain gauges in Fig. S5? From differencing the slip measurements in Fig. 2? At several points along the rupture front, or only an average from start to finish? In any case, it seems that it should be possible to estimate the velocity at several points along the rupture path for each event; how constant are those velocities? In Figs. 3 and 4a, each rupture plots at a single velocity with no error bar. I think that a new Supplementary Fig that shows velocity as a function of propagation distance for an assortment of ruptures would let the reader judge how constant the rupture velocities are, and what plausible error bars are.

Response: Thanks. We measure the rupture speed both from (1) the evolution of the slip distribution along the fault obtained from the optical measurements and further check the results against (2) shear strain observations. Following your suggestion, we added a section **Determination of rupture velocity** in **Methods** as follows to show the determination of rupture speed. We also added **Supplementary Figs. 5 and 6** to show the rupture velocity as a function of propagation distance for the events in **Fig. 2** and another three events. Finally, we added error bars

for rupture speeds in Fig. 4a.

**Determination of rupture velocity**

Taking advantage of the simultaneous measurements of the full-field displacement and local
shear strains, we can determine the rupture speeds by two independent methods. First, the full-
field DIC displacement measurement captures the slip distribution along the fault at different
instants in time. The rupture front is identified as the point where the fault-parallel particle
velocity \dot{u} exceeds a threshold \dot{u}_{th} . We first calculate the full-field fault-parallel particle velocity
\dot{u} by temporal differentiation at points along the fault, at an interval of ~ 0.6 mm (=1 pixels)
(Supplementary Fig. 5). Since the error in displacement is ~ 2 μm , the error in particle velocity is
~ 0.1 m/s (= 2 μm / 20 μs). We set $\dot{u}_{th} = 0.2$ m/s as the threshold for the passage of rupture front,
allowing the arrival time of the rupture to be determined. Because the slip velocity increases
sharply as the rupture passes, the arrival time is insensitive to the small change in \dot{u}_{th} . Finally,
we can obtain the average rupture velocity within 20 μs by temporal differentiation. Thus, the
error in rupture velocity is less than 30 m/s.

Second, from the local shear strain measurement at seven locations along the fault, the
arrival time of the rupture front can be picked from the shear strain record with an error of only 1
375 μs . The rupture velocity between neighboring strain gauge arrays is computed as the interval (80
376 mm) divided by the difference in arrival time. For supershear ruptures propagating at velocities
of ~ 2 km/s, the difference in arrival time is 40 μs . Thus, the relative error in rupture velocity is
2.5% (~ 50 m/s); for subRayleigh ruptures propagating at a velocity of ~ 1 km/s, the error is
smaller (~ 25 m/s).

The along-fault rupture velocity calculated from these two independent methods is shown in
Supplementary Figs. 5 and 6. These independently evaluated velocities match closely and with
nearly constant velocity - as there is no sharp accelerations or decelerations. The variation in
rupture velocity is small and usually within ~ 0.1 km/s – likely resulting from small variations in
applied fault stress (Supplementary Fig. 4). The average rupture velocity is computed by
averaging the local rupture velocity along strike.

Supplementary Figure 5. Three events with different rupture speeds elucidate the rupture dynamics of laboratory earthquakes. **a, b, c**, Spatio-temporal evolution of fault slip and evolution of local shear stress for **(a)** subRayleigh **(b)** subEshelby **(b)** and **(c)** supershear ruptures. **d, e, f**, Spatio-temporal evolution of fault-parallel slip velocity and evolution of local shear stress for the same events as in **a, b**, and **c**. **g, h, i**, Slip profile validates the crack-like sustained rupture for the same events as in **a, b**, and **c**. **j, k, l**, Distribution of initial shear stress τ_0 and dynamics stress drop for the same events as in **a, b**, and **c**. **m, n, o**, Rupture velocity along the fault for the same events as in **a, b**, and **c**. The nucleation zone is defined by the red rectangle.

Supplementary Figure 6. Evolution of rupture velocity along the fault for events shown in Fig. 2. The rupture velocity is derived from the full-field measurement of fault slip and local shear strain evolution. The nucleation zone is defined by the red rectangle. For the events with low resolution in slip velocity, only the rupture speed derived from stain signals is presented.

#2a Figures 1b-e show some interesting trends. In all cases except the rough fault with $\sigma_x = 1$ and 2 MPa, the max and min values of $\sigma_y - \sigma_x$ seem to increase monotonically with time or event number. In Fig. 1e, the first ruptures on the “extremely smooth” faults at the different σ_N seem to share a common coefficient of friction of ~ 0.4 . It looks like the same might be said of the smooth fault, but it’s hard to tell. What is most surprising to me is that each color in Fig. 1e (each value of σ_x) follows a very nearly linear trend, with an intercept along the σ_p axis (zero shear strength at a finite normal stress) that seems (coincidentally?) to be very close to σ_x .

What does this say about the evolution of the fault surface with successive ruptures? Or could it suggest that, with successive ruptures, the stress along the fault departs from the simple prediction based on the assumed uniform applied load? My understanding is that the shear and normal stress here are determined from the applied loads; can this be checked using the near-fault strain gauges?

Response: We appreciate these insightful observations on the interesting data trend that we missed. We interpret the trend as follows.

In our experiments, the shear stress and normal stress resolved along the fault plane can be written as:

$$\tau = \frac{1}{2}(\sigma_Y - \sigma_x) \quad (R1)$$

$$\sigma_n = \frac{1}{2}(\sigma_Y + \sigma_x) \quad (R2)$$

(R1) + (R2) yields

$$\tau = \sigma_n - \sigma_x \quad (R3)$$

From equation (R3), there is a linear relation between σ_n and τ , which is exactly what you observed in **Fig.1e**. The intercept on the σ_p axis is exactly σ_x (by taking $\tau=0$ in equation R3), which is consistent with your observation.

We agree that the fault surface may evolve with successive ruptures. It can be inferred from the observation that there is a minor increase in peak τ ($=(\sigma_y - \sigma_x)/2$) with successive ruptures, especially for the extremely smooth fault.

#2b In any case, I stared at this figure for a while, wondering how to interpret the x-intercept for each σ_x , before I realized that for the first rupture of a particular fault at each normal stress, the coefficient of friction was well defined. To avoid giving other readers this headache, it would be good to point this out. Although, now that I write this, was there a single manufacturing of each roughness? So that, for example, the same extremely smooth fault was successively subjected to $\sigma_x = 1$ MPa, 2 MPa, and 3 MPa? Then to explain the constant coefficient of friction for this fault for the first events at each normal stress, one would have to imagine that the surface was “reset” somehow (by separating and realigning the plates?). Or might this be evidence of a change in the way the applied loads are transferred to the fault surface? But then this would call into question the relation between rupture speed and $(\Delta\tau_d)^2/\sigma_N$, because presumably the different values of these parameters come from the stress drops in Fig. 1 that grow with successive rupture events.

Response: Again, we appreciate this thoughtful and insightful comment. Yes, you are right. For the first rupture of a particular fault at each normal stress, the coefficient of friction was well defined. This is indeed because every time we reset the fault roughness by refreshing the fault plane consistently by re-polishing.

#3 The model proposed here assumes that inelastic energy dissipation increases linearly with

rupture length. For natural faults, grain-boundary and larger cracks off the fault plane can act as seeds for this deformation. Is it possible to look for signs of permanent off-fault deformation in the PMMA using photoelasticity? Maybe even before and after the first event, to check for damage from the original cutting process, as well as the potential growth of the zone with propagation distance? Or, given the estimated size of the cohesive zone ($d/a \sim 0.04$) and the slip at a distance d behind the rupture front, estimate the near-tip strains and compare this to a plastic failure strength for PMMA? The latter would be much simpler, but still a nice addition.

Response: Thanks for the insightful suggestion. As shown in previous works, increasing energy dissipation with rupture length is a necessary condition for constant rupture speed on simple faults (Andrews, 2004; Templeton, 2009). Our assumption is thus a theoretical inference of this observation.

You suggested a few interesting methods to check this assumption. Because the “specific fracture energy” may include energy consumed at the rupture tip (fracture energy) (Svetlizky and Fineberg, 2014; Svetlizky et al., 2017), the off-fault yielding, and spatially extended dissipation (Brenner and Bouchbinder, 2021), any direct quantification of this is difficult given the constraints in our model – and would require a substantial plan and work. We hope that you agree that this is out of the scope of the current study. However, it can be an intriguing question for follow-up investigations.

Minor points:

1. I don't recall that it was ever mentioned if the ruptures here were crack-like or pulse-like. From the colors in Fig. 2 it appears that they might be pulse-like, but I suspect that is because the slip contours there saturate at $60 \mu\text{m}$. The stress histories in Fig. S5 don't obviously show an increase that would be associated with a pulse, but I'm not used to thinking about super-shear slip pulse stress histories. I think the equations in the Methods assume crack-like behavior. Please state whether these ruptures are pulse-like or crack-like, or a mixture; that information must exist.

Response: Thanks. In Fig. 2, to highlight the preslip in the nucleation phase, we set the maximum value corresponding the saturating red in the color bar as $60 \mu\text{m}$. Thus, it appears as though the ruptures are pulse-like. In fact, the ruptures are crack-like. In the revision, we added

Supplementary Fig. 5 to provide evidence to confirm to the reader the crack-like form of the rupture.

- line 37: “Understanding the controls on rupture speed is thus crucial for the earthquake source mechanisms and thus assessment of seismic hazards.” The “crucial for the earthquake source mechanisms” seems awkward here, and I’m not really sure what it means. How about “crucial for understanding earthquakes and thus ...”?

Response: We appreciate this helpful comment and revise the manuscript accordingly. We revise this sentence as:

38 damage due to the passage of the Mach wave ^{1,10,11}. **The controls on rupture speed is thus crucial**
39 **for understanding earthquakes** and **the attendant** assessment of seismic hazards.

- line 53: “the Burridge-Andrews mechanism”. For a paper in a general journal this assumes too much knowledge on the part of the reader. Something like “the Burridge-Andrews mechanism, where the jump to super-shear speeds is achieved through the growth of a separate rupture at the shear wave front ahead of the main rupture”, or a separate sentence if that makes the sentence too long. You can probably come up with something better.

Response: We appreciate this helpful comment and revise the manuscript accordingly. We revise this sentence in the revision as:

speed remains an open question. Theoretical studies ^{21,22} and laboratory experiments ²³ have
shown that such a rupture often gradually accelerates to Rayleigh speed and that some
subRayleigh ruptures can jump over the forbidden zone to a supershear speed through the
Burridge-Andrews mechanism - **where the jump to supershear speeds is achieved through the**
**growth of a separate rupture at the shear wave front ahead of the main rupture. Alternatively,**
**rupture can also** directly accelerate through this forbidden zone into the supershear region ²⁴.

- line 55: “at the onset”. Presumably there is a length scale below which the observations cannot

reach. Something like “from the first moments that a velocity can be discerned” seems safer.

Response: Thanks. We revise this sentence as:

58 However, the 2018 Mw 7.5 Palu earthquake attains a supershear speed at the **onset (or from the**
59 **first moments that a velocity can be discerned)**^{14,25}, challenging such conventional
60 understanding on supershear nucleation. The controls for rupture speeds represent another

5. line 61: “observations from natural earthquakes indicate that rupture speed may actually decrease with an increase in the stress drop.” If these observations pertain to super-shear as well as sub-Rayleigh ruptures, this should be stated explicitly.

Response: Thanks. These observations are mainly from sub-Rayleigh ruptures because supershear ruptures are relatively rare in natural earthquakes. We clarify this in the revision. Because we cannot explain the negative correlation, we move this sentence into the **Discussion** as suggested by **Referee # 1**. We revise this sentence as:

results. **We also note that some observations from natural earthquakes indicate that rupture speed**
**may decrease with an increase in the stress drop, especially for earthquakes with sub-Rayleigh**
**rupture speeds**^{60,61}. However, the validity of this finding may suffer from tradeoffs and
**uncertainties in estimating source properties for natural earthquakes.**

6. line 149: When I see “Stress ratio” here, I think of the Burridge-Andrews S, the ratio of the stress differences (peak shear stress at failure minus the ambient shear stress) to (ambient shear stress minus sliding friction), $(\tau_p - \tau_0)/(\tau_0 - \tau_f)$. But here, on line 153, “stress ratio” is identified with f_s , the ratio of the ambient shear stress to normal stress at the onset of slip, or just the static friction. Why not just say “Static friction and shear stress drop” on line 149?

Response: Thanks. We revise this sentence in the revision as:

**Factors governing rupture speed. The ratio of shear stress to normal stress**²⁶ and shear stress
drop^{29,45} have been proposed as the key factors that control the rupture speed. To explore the

7. line 155: “However, we observe that the specific relationship between τ_p/σ_p and V_r also varies with σ_x .” σ_x might be the experimentally-controlled variable here, with 3 discrete values, but it really might be σ_p that matters. Have you tried color-coding this figure by the value of σ_p ?

Response: We appreciate this insightful comment. We agree that σ_x is a well-controlled variable - hence we plot **Fig. 3** as in this manner. As you suggest, we color-code this figure using the value of σ_p as shown in the figure below. This figure is similar to **Fig. 3a**, since normal stress is correlated with σ_x as ($\sigma_n = \frac{1}{2}(\sigma_Y + \sigma_x)$).

Fig. R2. Dependence of rupture speed on fault roughness, the ratio of global peak shear stress to normal stress (τ_p/σ_p), and normal stress σ_p .

8. line 168: $\Delta\tau_d$ is defined as the “dynamic stress drop”. Is that the peak stress at the rupture front minus the quasi-steady sliding friction? Or the ambient shear stress minus the sliding friction? The inset to Fig. 4a suggests that it is the ambient stress minus the sliding friction, in that the small increase in stress just prior to the onset of weakening is ignored (I’m interpreting this increase as due to the arrival of the rupture front). But then how is this different from $\Delta\tau_s$?

Response: Thanks. We confirm that τ_d is defined as the “dynamic stress drop”, which is the difference between the ambient stress and the sliding friction - two well-defined values in the curve. Static stress drop is defined as the global stress drop in the macroscopic shear stress resolved on the fault (inset in **Fig.1f**). Static stress drop may involve the stress change caused by multiple ruptures whereas the dynamic stress drop quantifies the stress drop accompanying the first rupture.

In addition, we add a section in the Methods clarifying the definition of $\Delta\tau_d$ and $\Delta\tau_s$, and the difference between $\Delta\tau_d$ and $\Delta\tau_s$ as in the response to **Comment # 6** by **Referee #1** in the revision as:

**Static and dynamic stress drop**

In this work, static stress drop $\Delta\tau_s$ is defined as the global drop in the macroscopic shear
stress resolved on the fault (inset in Fig. 1f) - similar to previous studies⁶⁷. However, static stress
drop may not accurately quantify the change in shear stress associated with the rupture. This is
because static stress drop includes the contributions from both the specimen and the load frame
“machine stiffness” during unloading^{37,58}. Furthermore, static stress drop is an integration of
stress change resulting from multiple ruptures during a stick-slip event. The free boundary at the
end of the fault is subject to surface breakout when a rupture arrives³¹, potentially allowing
backward-propagating re-ruptures, as shown in Supplementary Fig. 5. Thus, the globally
measured static stress drop $\Delta\tau_s$ cannot represent the stress drop accompanying the first rupture
during a stick-slip event. To isolate the stress change of the first rupture from the global value,
we measure the dynamic stress drop $\Delta\tau_d$ of the first rupture from the evolution of the shear
stress, as shown in the inset in Fig. 4a. This treatment is in accordance with previous studies^{30,37}.

For natural earthquakes, because the faults are embedded in a nearly infinite medium (the
crust), boundary-induced repeated ruptures are likely absent. Thus, the dynamic stress drop $\Delta\tau_d$
of laboratory earthquakes is equivalent to the “seismological” stress drop $\Delta\tau$ in the investigation
of natural earthquakes, especially for ruptures on simple faults.

9. line 279: “It can be seen that a rougher fault has a higher β value”. What is β here? Do you mean α ?

Response: Thanks. We revise this sentence in the revision as:

319 smooth, and rough fault, respectively. It can be seen that a rougher fault has a higher α value,
320 consistent with existing studies⁵⁶.

10. line 323: k is defined on line 359, but it should be defined here.

Response: Thanks. We revise this sentence in the revision as:

for a static mode II rupture with $\Delta\tau_d$ being the dynamic stress drop, k the ratio between C_S and C_P
for plane stress conditions, a the crack half length, and μ the shear modulus.

11. line 356: “equation 6.6.92” should read 6.9.92.

Response: We appreciate this comment and correct the equation number accordingly.

For a subRayleigh rupture, the analytical expression for $g_{II}(\beta)$ is (equation 6.9.92 in reference
⁵¹)

References:

- Andrews, D. J. (2004), Rupture Models with Dynamically Determined Breakdown Displacement. *Bulletin Of The Seismological Society Of America*, 94(3), 769–775.
- Brener, E. A., & Bouchbinder, E. (2021), Unconventional singularities and energy balance in frictional rupture. *Nature Communications*, 12(1), 2585.
- Svetlizky, I., & Fineberg, J. (2014), Classical shear cracks drive the onset of dry frictional motion. *Nature*, 509(7499), 205-208.
- Svetlizky, I., Kammer, D. S., Bayart, E., Cohen, G., & Fineberg, J. (2017), Brittle Fracture Theory Predicts the Equation of Motion of Frictional Rupture Fronts. *Physical Review Letters*, 118(12), 125501.
- Templeton, E., 2009. Effects of Inelastic Off-Fault Deformation on the Dynamics of Earthquake Rupture and Branch Fault Activation. Harvard University.

Reviewer #3 (Remarks to the Author):

In this manuscript, the authors report an experimental study of interfacial ruptures propagating along a frictional interface at all velocities, from sub-Rayleigh to supershear. The main experimental results are the observation of the forbidden zone for rupture velocity, between the Rayleigh wave velocity and 1.08 of the shear wave velocity, ruptures propagating at constant velocity, and the instantaneous nucleation of shear ruptures without initiation by a sub-Rayleigh rupture. The authors develop an ad-hoc model based on self-similar crack propagation to describe their observations. They are also able to fit seismic data with their model.

The experimental study of the evolution of the velocity of frictional failures is not new and has been solved before for a similar system. However, the observations made in this study are different from those in the literature. In particular, the existence of a band gap has never been explicitly demonstrated on the basis of experimental data. I have some major concerns that I detail below regarding the validity of the observations and measurements. If the authors can convince me of the validity of their observations by providing additional graphs, then the manuscript might be worth publishing in Nature Communications, in my opinion.

Response: Yes, thanks, we appreciate this general comment. You succinctly summarize the main findings of our study. In particular, you identify the “band gap” in the rupture velocity as a new experimental finding. To our best knowledge, the *direct observation of a “steady” rupture speed other than the limiting speeds (i.e., P wave speed in the supershear regime and Rayleigh wave speed in the subRayleigh regime)* also constitutes a new finding. In the existing literature, the observed “steady” rupture speeds are usually the limiting speeds (Svetlizky et al., 2020; Xia et al., 2004; Xu et al., 2018). However, these ruptures may still be mildly transient because near the limiting speeds, the acceleration cannot be too large.

As in the responses to **Referee #1**, we attribute the observation of “steady” speeds to the stress uniformity along the fault in the current study. We have added **Supplementary Figs. 4-6** to validate our observations and support our claims. We trust that our responses and modifications have properly addressed your concerns on the analysis and interpretation of our results.

Major comments:

The dynamics of a frictional rupture has been considered in recent experimental papers (Svetlizky et al [29] then Passelègue et al [26] for sub-Rayleigh ruptures, Kammer et al [45] for supershear ruptures). The authors' claim is that in the present study, constant propagation velocities have been measured. I would like the authors to clarify some of their measurements and observations:

#1 The main result of the paper and the motivation for deriving a new model is that ruptures propagate at constant velocities. The reader should extract this observation from the space-time plots in Fig. 2. I am not convinced that the velocity of ruptures is indeed constant. I would like the authors to provide the velocity profiles as a function of space for the 9 experiments shown in Fig. 2.

Response: We appreciate this insightful and thoughtful comment. We added **Supplementary Figs. 5 and 6** to support our claim that the rupture velocity is nearly constant outside the nucleation zone as:

Supplementary Figure 5. Three events with different rupture speeds elucidate the rupture dynamics of laboratory earthquakes. a, b, c, Spatio-temporal evolution of fault slip and evolution of local shear stress for (a) subRayleigh (b) subEshelby (b) and (c) supershear ruptures. d, e, f, Spatio-temporal evolution of fault-parallel slip velocity and evolution of local shear stress for the same events as in a, b, and c. g, h, i, Slip profile validates the crack-like sustained rupture for the same events as in a, b, and c. j, k, l, Distribution of initial shear stress τ_0 and dynamics stress drop for the same events as in a, b, and c. m, n, o, Rupture velocity along the fault for the same events as in a, b, and c. The nucleation zone is defined by the red rectangle.

Supplementary Figure 6. Evolution of rupture velocity along the fault for events shown in Fig. 2. The rupture velocity is derived from the full-field measurement of fault slip and local shear strain evolution. The nucleation zone is defined by the red rectangle. For the events with low resolution in slip velocity, only the rupture speed derived from strain signals is presented.

#2 Assuming that the ruptures do indeed propagate at constant velocity, the stress distributions along the interface need to be clarified. The authors consider that the interface is uniformly loaded based on the fact that nucleation occurs anywhere along the interface. However, the measurements provided by strain gauges must explicitly establish this distribution. Indeed, a non-homogeneous stress distribution can lead to non-uniform rupture dynamics. One possible scenario is that the load decreases along the interface, leading to non-accelerating cracks. But this needs to be checked on the data. I would like the authors to provide the distribution of stresses, both normal and shear, as a function of space - along the interface - prior to rupture propagation.

Response: We appreciate this insightful comment. We added **Supplementary Fig. 4** to show the distribution of shear stress and dynamic stress drop along the fault in three sequences of stick-slip events. Given the large dimension of the specimen, the shear stress before the onset of rupture is relatively uniform, especially in the central segment of the laboratory fault where ruptures initiate and propagate. Also the sign of the stress gradient varies, excluding the interesting scenario you propose to explain the steady velocity. The distributions of shear stress and dynamic stress drop are clearly more uniform than those in direct shear experiments (Ben-David et al., 2010). What is

more interesting from the figure is that the distribution of dynamic stress drop is more uniform than the distribution of shear stress. We believe that the dynamic stress drop measurement is more reliable because it is transient and thus immunized from error induced by thermal drift. We should clarify that the local normal stress was not measured in this work because of the finite and limited number of channels available for data acquisition.

Supplementary Figure 4. Distribution of initial shear stress τ_0 and dynamic stress drop $\Delta\tau_d$ for three sequences of stick-slip events. a, Stick-slip events on extremely smooth fault with $\sigma_x = 2$ MPa. **b,** Stick-slip events on smooth fault with $\sigma_x = 2$ MPa. **c,** Stick-slip events on rough fault with $\sigma_x = 3$ MPa. Initial shear stress and dynamic stress drop are measured by strain gauges

(S1~S7) mounted along the fault. Initial shear stress is the value measured at about 1 ms before the onset of the instability. When the sample is not subjected to loading, the shear stress is considered as zero for reference. Different colors correspond to different event ID in the stick-slip sequence.

#3 From an experimental point of view, the system is made of 20mm-thick blocks. How do the authors ensure that the fracture fronts are straight? If this is not the case, the velocity established from measurements on one face of the blocks will correspond to an effective velocity different from the local and instantaneous velocity, e.g. Rubino et al (Nature 2022, DOI: 10.1038/s41586-022-04749-3). It would lead to a misestimating of the $1.08 C_s$ limit of the forbidden zone.

Response: We appreciate this insightful comment. This 3D effect may be present/prevalent in laboratory earthquakes using thick plates. However, such an effect would dominate only in the initial stage before the rupture penetrates the entire fault width (Rubino et al., 2022), for 2D plane stress faults. In this case, the rupture front can be assumed as a straight line parallel to the fault width (Xu et al., 2018). In this work, we focus on the rupture propagation outside the nucleation zone. Besides, the rupture velocity is measured on the front and back surfaces of the blocks simultaneously using both DIC and strain gauges. The consistency of rupture speed measurement results (**Supplementary Fig. 6**) further justifies our assumption.

#4 Based on these observations, the authors establish that the LEFM equation of motion is not valid in their experiment. In this paragraph (from 1.166), the paper by Passelègue et al [26] is mentioned as establishing the criterion from the LEFM, while it was originally derived and experimentally validated for a 2D system in Svetlizky et al [29]. The explicit citation should be corrected.

Response: We appreciate this insightful comment. We note that the rupture in previous experiments (Passelègue et al., 2020) is two-dimensional (2D), involving mode II and mode III rupture whereas the adopted theoretical model is suitable for one-dimensional rupture. We agree that the original criterion was established by Svetlizky et al., (2017). Thus, we revised the citation on as:

Within the framework of linear elastic fracture mechanics, rupture propagation is governed
by the energy balance ⁴⁷. According to this rule, a theoretical model has been derived for
subRayleigh ruptures, where the rupture speed is correlated with $\Delta\tau_d^2/\sigma_N$, with $\Delta\tau_d$ and σ_N
being the dynamic stress drop and the normal stress, respectively ²⁸. This useful conclusion

#5 A model is then derived from the self-similar crack model to obtain a new equation of motion of a crack. The derivation appears valid, however, strong assumptions must be made:

(i) Increase in fracture energy with fracture length: the authors validate this effect on the basis of seismological data. However, the energy dissipated during a seismic event (G_{EQ}) and the fracture energy used in fracture mechanics (G_{FM}) are not the same quantities. Ke et al (Nat. Comm. 2022, DOI: 10.1038/s41467-022-28647-4) recently showed that by holding G_{FM} constant, they find an increase in G_{EQ} with the rupture length.

(ii) Increase in fracture energy with rupture speed: the material tested here is PMMA, as in [29] where no strong dependence of fracture energy on velocity was detected. The references cited on line 331 do not seem to refer to PMMA and to my knowledge there is no such effect in PMMA reported in the literature.

Response: (i) Thanks. In the original manuscript, the “fracture energy” refers to total energy dissipation, which is a lumped representation of the off-fault dissipation and non-edge-localized dissipation. To avoid confusion, we have renamed the total energy dissipation as “**specific fracture energy**”. This energy is equivalent to the “seismological” fracture energy (or breakdown work). Thus, our results do not contradict those of *Ke et al.* (Nat. Comm. 2022). We add clarification in the revision as:

and supershear ruptures. The specific fracture energy Γ_{II} can be interpreted as the energy not only
consumed at the rupture tip (fracture energy), but also the inelastic and spatially extended
dissipation ⁵². Experiments have revealed that specific fracture energy may be a function of the
rupture speed ⁵³ and rupture length ⁵⁴, at least for mode I ruptures. We assume that the specific

and as:

In a self-similar rupture model, an inherent theoretical consequence is that the specific
fracture energy Γ_{II} must increase with the rupture length⁷². Γ_{II} is a generalized concept that may
contain energy consumed at the rupture tip (fracture energy), inelastic dissipation and spatially
extended dissipation⁵². The proportionality between Γ_{II} and rupture length is consistent with
dynamic rupture models with off-fault dissipation^{49,52,72} and non-edge-localized dissipation⁶⁸.
Also, this proportionality is consistent with field observations that “seismological” fracture
energy (breakdown work) scales with the rupture size⁷³.

(ii) We appreciate this insightful comment. We assume that the energy dissipation is rupture velocity-dependent without specifying the exact functional form. The functional form is obtained by fitting the experimental data (**Supplementary Fig. 7**). Interestingly, this figure shows that for sub-Rayleigh ruptures with speeds lower than 0.4β , $f(\beta)$ is nearly constant. This result is consistent with reference [29], as you kindly identify, for the same range of rupture speeds.

#6 More generally, in [29] the equation of motion of a crack with constant fracture energy describes the data for all ruptures up to the Rayleigh wave speed, without a need for introducing a dependence both on length and velocity. Can the authors comment on the origin of the differences in behavior in these two very similar systems?

Response: We appreciate this insightful comment. We respectfully note, that in the work of Svetlizky et al. (2017), the estimated fracture energy varies by 40% along the fault. It is possible that the effect of rupture speed on fracture energy is hidden in the fluctuation in the fracture energy. Conversely, constant energy dissipation cannot explain the constant rupture speed (Templeton, 2009; Weng and Ampuero, 2020). Therefore, rupture length-dependent fracture energy (Passelègue et al., 2020) or rupture speed-dependent fracture energy (Weng and Ampuero, 2022) are proposed as mechanisms to interpret constant or steady rupture speed.

Furthermore, we realized during revision that it is not the “**fracture energy**” but the “**specific fracture energy**” that depends on rupture length and rupture speed. The specific fracture energy is a combined sum of fracture energy, the off-fault yielding, and spatially extended dissipation. Fracture energy may only constitute a small portion of the specific fracture energy and thus the

length-dependent specific fracture energy reflects the behavior of other dissipations (e.g., (Andrews, 2004; Templeton, 2009)).

#7 In addition to these two assumptions, the model involves rewriting the expression for $g_{II}(v)$ for a self-similar crack instead of a steady crack. However, the two expressions are very similar especially for intermediate velocities. I would like the authors to provide (i) a quantitative comparison of the results from the two models – self-similar and steady, and (ii) a comparison of their measurements with the model derived for a steady rupture.

Response: We appreciate this comment. We agree that the two expressions are very similar. The comparison between the universal function $g(v)$ and the specific function for self-similar rupture $w(v)$ can be found in **Section 4.3** in (Templeton, 2009) as reproduced in **Fig. R2**. It appears as if the difference between these two functions is minor. However, we prefer to use the specific function $w(v)$ out of concern for theoretical rigor.

Fig. R2. Comparison between the universal functions $g(v)$ and specific function of self-similar rupture $w(v)$ for different fracture modes (Templeton, 2009).

#8 Eventually, the logic of the study seems to me to be flawed in one respect. The authors derive a different model from LEFM because it cannot predict constant velocity propagation with constant fracture energy. However, later the authors consider a non-constant fracture energy to describe their data (Eqs. 4 & $D_c = \eta a$). With these assumptions, LEFM could describe the data as well as the self-similar crack.

Response: Thanks. We first clarify our rationale by noting the differences between LEFM or cohesive zone models against steady-state or self-similar models. LEFM is concerned with the material response around the crack-tip. In this model, the material is assumed to be elastic except for a vanishingly small region at the crack-tip (Anderson, 2005). An alternative theoretical model to LEFM is cohesive zone models (Barenblatt, 1962; Dugdale, 1960), where the inelastic material response is assumed to be restricted to a process zone ahead of the crack-tip. Conversely, steady-state or self-similar models are concerned with the evolution of mechanical fields with respect to a crack-tip moving at constant speed (Broberg, 1999). The fracture energy is a constant in the steady-state model, but increases with crack length in the self-similar model.

Thus, we adopt a different model (i.e., self-similar) from “steady-state” (not LEFM) because it cannot predict constant velocity propagation with constant fracture energy. The non-constant fracture energy assumption is made following the literature. In fact, we follow the classical self-similar solutions of Broberg (1999), who adopted the LEFM theoretical framework in the sub-Rayleigh regime but a cohesive zone model in the supershear regime as detailed in **Methods**.

Therefore, LEFM can accommodate either steady-state or self-similar dynamic fracture growth. Similarly, cohesive zone models can also accommodate either steady-state or self-similar dynamic fracture growth. Thus, there is no contradiction between the use of LEFM and self-similar models.

#9 The paper extends the model to supershear fractures. How does it compare to other more realistic than self-similar crack models, such as the solution of Huang and Gao (JAM 2001, DOI: 10.1115/1.1357871)?

Response: Both Broberg (1999) and Huang and Gao (JAM 2001, DOI: 10.1115/1.1357871 (Huang and Gao, 2001)) assume a Barenblatt-type process region, but differ in the form of the stress distribution within the process zone. These different forms do not significantly affect the general results, which is reflected in the similar shape of the energy release rate as a function of rupture speed (**Fig.1** in Huang and Gao (2001) and **Fig.6.9.12** in Broberg (1999)).

#10 The authors report qualitative observations on the effect of surface roughness on interface stability and strength. Can they provide an interpretation of these observations?

Response: Thanks. We add an interpretation in the **Discussion** in the revised manuscript as:

**Discussion**

Mature faults with large slip offsets are typically smoother than short-offset immature faults
⁵⁵. Congruently, high rupture speed is correlated with mature fault sections ⁵⁶. This is
qualitatively consistent with previous experimental observations that increasing fault roughness
has a stabilizing effect on fault rupture ⁵⁷ - seemingly contrary to our findings, here. This
discrepancy may arise from the limited range of roughness involved in this study. Recently, the
effect of fault roughness on rupture dynamics has been resolved over a wide range of
roughnesses ⁵⁸. This has defined a critical roughness, below which the stress drop increases with
roughness, and beyond which the stress drop decreases with roughness. This suggests that an
increase in roughness may not necessarily stabilize fault rupture. In this work, the fault is overall
smooth (Supplementary Fig. 1). Thus, an increase in fault roughness may promote fault
instability according to observations by Morad, et al. ⁵⁸. Our results are generally consistent with
those of Xu, et al. ³¹, who demonstrated that rough faults could indeed accommodate supershear
rupture.

Minor comments:

1. lines 67-68: “The rupture speed is dictated by the driving load that depends on the dynamic stress drop and the normal stress”. The stress drop depends on the driving load and not the opposite.

Response: Thanks. In this sentence, “driving load” has a specific meaning. It refers to $\frac{\Delta\tau_d^2}{\sigma_N}$ as defined later in the main text. As now noted in the revision as:

rupture length by Passelègue, et al ²⁸, we arrive at a unique relation between $\Delta\tau_d^2/\sigma_N$ and β ,
with the former being the generalized driving load and the latter being the rupture speed
normalized by the P wave speed. This theoretical result nicely confirms the experimental

2. line 89: The definition of the peak shear stress and of the static shear stress drop should be clarified and defined on the inset of Fig. 4a, as $\Delta\tau_d$. Does it correspond to the initial

stress applied to the fault when the rupture nucleates or to the contact shear strength? The latter cannot be measured directly.

Response: Thanks. The definitions of peak shear stress and static shear stress drop are as illustrated in the inset of Fig. 1f. The peak shear stress is defined as the global shear strength of the fault, which is measured from the load cell at a sampling rate of 100 Hz.

Fig. 1 Stick-slip events mimicking natural earthquakes. **a**, Schematic of laboratory earthquake model. The laboratory fault, with total length of 650 mm and observation window of 550 mm, is subjected to biaxial loading. The fault slip directions are indicated by red arrows. **b-d**, Differential stress ($\sigma_y - \sigma_x$) as a function of time at increasing horizontal stress ($\sigma_x = 1, 2,$ and 3 MPa) on extremely smooth, smooth, then rough faults. **e**, Stress state, global peak shear stress τ_p as a function of peak normal stress σ_p , at the onset of instability for each event. Black dashed lines correspond to different ratios of τ_p/σ_p . **f**, Static shear stress drop $\Delta\tau_s$ as a function of peak normal stress τ_p for each event. Black dashed lines indicate different values of relative shear stress drop $\Delta\tau_s/\tau_p$. The inset illustrates the definition of static shear stress drop $\Delta\tau_s$ and peak normal stress τ_p .

3. lines 183-184: The writing is confusing. “In this model fracture speed is constant... The fracture energy is thus constant in this model”... “an expanding fracture has to accelerate”. It should be corrected.

Response: We appreciate this comment and revise the manuscript as:

**Self-similar analysis.** Steady-state fracture models have been widely used to analyze dynamic
rupture^{12,48}. In this model, for fractures with a given speed, the mechanical fields are assumed to
remain invariant with respect to the reference-frame of the moving crack-tip. Because the
mechanical fields (including the stress field) are invariant, the stress intensity factor remains the
same. Constant stress intensity factor and constant fracture velocity result in constant fracture
energy in this model. Cracks propagating in speed regimes $V_r < C_R$ and $C_S < V_r < \sqrt{2}C_S$ are shown to
be unstable, while the speed regime $\sqrt{2}C_S < V_r < \sqrt{2}C_P$ is stable^{12,13}. More generally, stable rupture
speed is not accessible for simple models with constant fracture energy^{20,49}. Therefore,

4. line 205: Note that speeds between C_R and $\sqrt{2}C_S$ have been measured in Svetlizky et al (JMPS 2020, DOI: 10.1016/j.jmps.2019.103826) and Kammer et al [45].

Response: We appreciate this comment and cite these references in the revision as:

mechanisms for supershear ruptures are possible³⁹⁻⁴¹. Furthermore, nearly steady sub-Eshelby
supershear speeds are observed (Supplementary Figs. 5 and 6), which confirms the other puzzle
in the 2018 Palu earthquake. Although rupture speeds between C_S and C_E have been reported
149^{38,42}, the observed rupture speeds are always transient. These observations of non-steady rupture
speed may be associated with the stress gradient along the fault in direct shear experiments²⁶ as
first pointed out by Scholz^{43,44}.

5. line 356: the reference to section 6.6.92 should read 6.9.92.

Response: We appreciate this comment and correct the equation number in the revision as:

For a sub-Rayleigh rupture, the analytical expression for $g_{II}(\beta)$ is (equation 6.9.92 in reference
459⁵¹)

References:

- Anderson, D. L., 2005. Fracture Mechanics. CRC Press, Boca Raton.
Andrews, D. J. (2004), Rupture Models with Dynamically Determined Breakdown Displacement. *Bulletin Of The Seismological Society Of America*, 94(3), 769–775.

- Barenblatt, G. I. (1962), The Mathematical Theory of Equilibrium Cracks in Brittle Fracture. *Advances In Applied Mechanics*, 7, 55-129.
- Ben-David, O., Cohen, G., & Fineberg, J. (2010), The dynamics of the onset of frictional slip. *Science*, 330(6001), 211-214.
- Broberg, K. B., 1999. Cracks and fracture. Academic Press, San Diego.
- Dugdale, D. S. (1960), Yielding of steel sheets containing slits. *Journal of the Mechanics and Physics of Solids*, 8, 100-104.
- Huang, Y., & Gao, H. (2001), Intersonic Crack Propagation— Part I: The Fundamental Solution. *Journal of Applied Mechanics*, 68, 169-175.
- Passelègue, F. X., Almakari, M., Dublanchet, P., Barras, F., Fortin, J., & Violay, M. (2020), Initial effective stress controls the nature of earthquakes. *Nature Communications*, 11, 5132.
- Rubino, V., Lapusta, N., & Rosakis, A. J. (2022), Intermittent lab earthquakes in dynamically weakening fault gouge. *Nature*, 606(7916), 922-929.
- Svetlizky, I., Albertini, G., Cohen, G., Kammer, D. S., & Fineberg, J. (2020), Dynamic fields at the tip of sub-Rayleigh and supershear frictional rupture fronts. *Journal of the Mechanics and Physics of Solids*, 137, 103826.
- Svetlizky, I., Kammer, D. S., Bayart, E., Cohen, G., & Fineberg, J. (2017), Brittle Fracture Theory Predicts the Equation of Motion of Frictional Rupture Fronts. *Physical Review Letters*, 118(12), 125501.
- Templeton, E., 2009. Effects of Inelastic Off-Fault Deformation on the Dynamics of Earthquake Rupture and Branch Fault Activation. Harvard University.
- Weng, H., & Ampuero, J.-P. (2020), Continuum of earthquake rupture speeds enabled by oblique slip. *Nature Geoscience*, 13(12), 817-821.
- Weng, H., & Ampuero, J. P. (2022), Integrated rupture mechanics for slow slip events and earthquakes. *Nature Communications*, 13(1), 7327.
- Xia, K., Rosakis, A. J., & Kanamori, H. (2004), Laboratory earthquakes: The sub-Rayleigh-to-supershear rupture transition. *Science*, 303(5665), 1859-1861.
- Xu, S., Fukuyama, E., Yamashita, F., Mizoguchi, K., Takizawa, S., & Kawakata, H. (2018), Strain rate effect on fault slip and rupture evolution: Insight from meter-scale rock friction experiments. *Tectonophysics*, 733, 209-231.

REVIEWERS' COMMENTS

Reviewer #1 (Remarks to the Author):

I have read the authors' replies to all three reviewers, as well as their revised texts designed to improve the clarity of the manuscript. Overall, I am quite satisfied with the current version, especially the confirmations of (i) nearly constant rupture speed from two independent measurement methods and (ii) the relatively uniform distributions of initial stress and stress drop. I think the manuscript can be readily accepted after the following minor points are addressed.

(1) First of all, I can confirm that the authors have used an updated value of P wave velocity under plane stress conditions (Figs. 3 and 4, Supplementary Fig. 7). It is nice to see that, after this update, the upper limit of rupture speed in the supershear regime can be clustered more closely at C_p (Figs. 3 and 4a) than in the previous version. I can also confirm that the fitting formulae (lines 469 and 498) have been slightly revised, in accordance to the updated P wave velocity. On the other hand, I am not sure if the inclined reference line for C_p in Fig. 2 and Supplementary Fig. 5 have been updated. Would the authors have a check?

(2) In the revised version, the authors have invoked the non-edge-localized dissipation mechanism (Brener and Bouchbinder, 2021) as one possible source for scale-dependent specific fracture energy. While this is fine for a general discussion, I would like the authors to pay attention to the following point: if the non-edge-localized dissipation does apply, then the criterion for reading dynamic stress drop would need to be updated, by taking into account the long tail of the slip-weakening curve. In the current work, the authors mainly read dynamic stress drop near the rupture front (inset in Fig. 4a), so that the most relevant mechanism to invoke, in my opinion, would be off-fault plastic yielding near the rupture front. This issue does not influence the general discussion, but it would be better to clarify it somewhere in the Methods.

Main text:

Line 107: "peak normal stress" => "peak shear stress"

Line 201: remove $\sqrt{2}$ right before C_p

Supplementary Information:

Line 62: "(b) and (c)" should be "and (c)"

Lines 65-66: "dynamics stress drop" => "dynamic stress drop"

Line 76: "stain" => "strain"

Reviewer #2 (Remarks to the Author):

I think the authors have responded to all my comments adequately. I think the new Supplementary Figures 4 and 5, showing the propagation speed as a function of location, make this a stronger paper. The experiments are very noteworthy and should be of interest to a wide audience, so I think this paper is suitable for publication in Nature Communications as is.

Reviewer #3 (Remarks to the Author):

The authors' response effectively reinforces their observations, making them more convincing to the reader. While I may not be entirely persuaded by all the reasons they give to explain the discrepancies with other studies, I believe that this study adds significant value to the existing literature and expands the range of observations made on laboratory faults. Therefore, I recommend the revised version of the manuscript be published in Nature Communications.

Below I report my concerns regarding the authors' arguments, and suggest some minor improvements to the manuscript.

- Regarding the need of using a self-similar crack model, I thank the authors for their response. In the end they decided to use a self-similar model, but LEFM with velocity dependent fracture energy

could work as well. The way the manuscript is written lets us understand that a self-similar model is needed to describe the observations. I suggest to clarify this for the reader and for the scientific sound, I suggest to try LEFM with non-constant fracture energy.

- The authors report observing steady fractures, whereas Fineberg's group reports observing accelerating fractures. The authors attempt to explain this discrepancy by claiming that Fineberg's stress distribution is not uniform, citing a 2010 paper. However, in more recent publications, the torque is actually used to generate nearly uniform stress distribution in the central part of the fault (Svetlizky and Fineberg 2014 and followings). Furthermore, upon examination of Supplementary Fig. 5, it appears that the stress distribution is not significantly more homogeneous than in the above-mentioned experiments, except at the corner. Thus, I would recommend taking this particular argument with a grain of salt.

- The authors have corrected the fracture energy term by using the specific fracture energy. While I agree that these two quantities are distinct, I do not see a compelling reason to use the specific fracture energy in the rupture speed selection model. The additional dissipation terms included in the specific fracture energy are typically generated behind the rupture tip and are not expected to affect the rupture speed prior to the first reflection.

- Supp. Fig. 5 provides support for the observation of constant rupture speed. However, I recommend narrowing the range of the plot, as the current range of 0 to c_p makes it difficult to accurately extract the velocity value from the plot.

- I thank the authors for the clarification of the stress drop definitions, which are now much clearer.

RESPONSE TO REVIEWERS' COMMENTS

Review comments are shown in plain text, responses in blue, and changes in red in the revised Manuscript.

Comments from reviewers

Reviewer #1 (Remarks to the Author):

I have read the authors' replies to all three reviewers, as well as their revised texts designed to improve the clarity of the manuscript. Overall, I am quite satisfied with the current version, especially the confirmations of (i) nearly constant rupture speed from two independent measurement methods and (ii) the relatively uniform distributions of initial stress and stress drop. I think the manuscript can be readily accepted after the following minor points are addressed.

Response: Great! Thanks. We are glad that you are satisfied with our responses and revision. We address your additional points below.

(1) First of all, I can confirm that the authors have used an updated value of P wave velocity under plane stress conditions (Figs. 3 and 4, Supplementary Fig. 7). It is nice to see that, after this update, the upper limit of rupture speed in the supershear regime can be clustered more closely at C_p (Figs. 3 and 4a) than in the previous version. I can also confirm that the fitting formulae (lines 469 and 498) have been slightly revised, in accordance to the updated P wave velocity. On the other hand, I am not sure if the inclined reference line for C_p in Fig. 2 and Supplementary Fig. 5 have been updated. Would the authors have a check?

Response: Thanks. Yes, we forgot to update the inclined reference line for C_p in Fig. 2 and Supplementary Fig. 5. We correct this problem in the revision.

(2) In the revised version, the authors have invoked the non-edge-localized dissipation mechanism (Brener and Bouchbinder, 2021) as one possible source for scale-dependent specific fracture energy. While this is fine for a general discussion, I would like the authors to pay attention to the following point: if the non-edge-localized dissipation does apply, then the criterion for reading dynamic stress drop would need to be updated, by taking into account the long tail of the slip-weakening curve. In the current work, the authors mainly read dynamic stress drop near the rupture front (inset in Fig. 4a), so that the most relevant mechanism to invoke, in my opinion, would be off-fault plastic yielding near the rupture front. This issue does not influence the general discussion,

but it would be better to clarify it somewhere in the Methods.

Response: We agree that the non-edge-localized dissipation is not compatible with the measurement of dynamic stress drop. Thus, we remove the non-edge-localized dissipation mechanism (Brener and Bouchbinder, 2021) in the revision. The sentence on Lines 423-425 now reads: “Several mechanisms have been proposed to provide this increase, including off-fault yielding⁵³, provision of specific friction laws^{70,71} and slip-rate-dependent fracture energy⁶³.”

Main text:

Line 107: “peak normal stress” => “peak shear stress”

Response: Revised.

Line 201: remove $\sqrt{2}$ right before C_p

Response: Revised.

Supplementary Information:

Line 62: “(b) and (c)” should be “and (c)”

Response: Revised.

Lines 65-66: “dynamics stress drop” => “dynamic stress drop”

Response: Revised.

Line 76: “stain” => “strain”

Response: Revised.

Reviewer #2 (Remarks to the Author):

I think the authors have responded to all my comments adequately. I think the new Supplementary Figures 4 and 5, showing the propagation speed as a function of location, make this a stronger paper. The experiments are very noteworthy and should be of interest to a wide audience, so I think this paper is suitable for publication in Nature Communications as is.

Response: Great! Thanks. We are glad these figures helped.

Reviewer #3 (Remarks to the Author):

The authors' response effectively reinforces their observations, making them more convincing to the reader. While I may not be entirely persuaded by all the reasons they give to explain the discrepancies with other studies, I believe that this study adds significant value to the existing

literature and expands the range of observations made on laboratory faults. Therefore, I recommend the revised version of the manuscript be published in Nature Communications.

Response: We are glad that you are satisfied with our responses and revision. We address your additional points below.

Below I report my concerns regarding the authors' arguments, and suggest some minor improvements to the manuscript.

- Regarding the need of using a self-similar crack model, I thank the authors for their response. In the end they decided to use a self-similar model, but LEFM with velocity dependent fracture energy could work as well. The way the manuscript is written lets us understand that a self-similar model is needed to describe the observations. I suggest to clarify this for the reader and for the scientific sound, I suggest to try LEFM with non-constant fracture energy.

Response: We agree that other mechanisms may also work. We have already cited several possible models, such as the velocity dependent fracture energy (Weng and Ampuero, 2022) and LEFM with non-constant fracture energy (Templeton, 2009), on Lines 423-425.

Following your suggestion to further clarify this point, we modify the sentence at the end of Discussion (Lines 304-307) as: “Because natural faults are more complex than laboratory faults, alternative models in the framework of linear elastic fracture mechanics coupled with velocity-dependent fracture energy⁶³ or non-constant fracture energy⁵³ may also work.”

- The authors report observing steady fractures, whereas Fineberg's group reports observing accelerating fractures. The authors attempt to explain this discrepancy by claiming that Fineberg's stress distribution is not uniform, citing a 2010 paper. However, in more recent publications, the torque is actually used to generate nearly uniform stress distribution in the central part of the fault (Svetlizky and Fineberg 2014 and followings). Furthermore, upon examination of Supplementary Fig. 5, it appears that the stress distribution is not significantly more homogeneous than in the above-mentioned experiments, except at the corner. Thus, I would recommend taking this particular argument with a grain of salt.

Response: We agree with your point. Following your suggestion, we add a sentence (Lines 160-162) in the discussion as: “If the rupture initiates from the edge of the direct shear specimen

that suffers the most from the stress nonuniformity, then such an issue may persist even if the stress is uniform in the central part of the fault ⁴⁵.”

- The authors have corrected the fracture energy term by using the specific fracture energy. While I agree that these two quantities are distinct, I do not see a compelling reason to use the specific fracture energy in the rupture speed selection model. The additional dissipation terms included in the specific fracture energy are typically generated behind the rupture tip and are not expected to affect the rupture speed prior to the first reflection.

Response: Agreed. Non-edge-localized dissipation should not be included in the specific fracture energy. We thus revise the sentence on Lines 231-232 as: “The specific fracture energy Γ_{II} can be interpreted as not only the fracture energy at the rupture tip, but also other tip-localized dissipation ⁵³.”

- Supp. Fig. 5 provides support for the observation of constant rupture speed. However, I recommend narrowing the range of the plot, as the current range of 0 to c_p makes it difficult to accurately extract the velocity value from the plot.

Response: Modified.

- I thank the authors for the clarification of the stress drop definitions, which are now much clearer.

Response: We are glad that you are satisfied with our revision.

References:

- Templeton, E., 2009. Effects of inelastic off-fault deformation on the dynamics of earthquake rupture and branch fault activation. Harvard University.
- Weng, H., & Ampuero, J. P. (2022), Integrated rupture mechanics for slow slip events and earthquakes. *Nature Communications*, 13(1), 7327.